# LLM-grounded Diffusion: Enhancing Prompt Understanding of Text-to-Image Diffusion Models with Large Language Models

**Long Lian**                                                            *longlian@berkeley.edu*
*UC Berkeley*

**Boyi Li**                                                              *boyili@berkeley.edu*
*UC Berkeley*

**Adam Yala**                                                           *yala@berkeley.edu*
*UC Berkeley, UCSF*

**Trevor Darrell**                                              *trevordarrell@berkeley.edu*
*UC Berkeley*

**Reviewed on OpenReview:** `https://openreview.net/forum?id=hFALpTb4fR`

## Abstract

Recent advancements in text-to-image diffusion models have yielded impressive results in generating realistic and diverse images. However, these models still struggle with complex prompts, such as those that involve numeracy and spatial reasoning. This work proposes to enhance prompt understanding capabilities in diffusion models. Our method leverages a pretrained large language model (LLM) for grounded generation in a novel two-stage process. In the first stage, the LLM generates a scene layout that comprises captioned bounding boxes from a given prompt describing the desired image. In the second stage, a novel controller guides an off-the-shelf diffusion model for layout-grounded image generation. Both stages utilize existing pretrained models without additional model parameter optimization. Our method significantly outperforms the base diffusion model and several strong baselines in accurately generating images according to prompts that require various capabilities, *doubling* the generation accuracy across four tasks on average. Furthermore, our method enables instruction-based multi-round scene specification and can handle prompts in languages not supported by the underlying diffusion model. We anticipate that our method will unleash users' creativity by accurately following more complex prompts. Our code, demo, and benchmark are available at: `https://llm-grounded-diffusion.github.io`.

## 1 Introduction

The field of text-to-image generation has witnessed significant advancements, particularly with the emergence of diffusion models. These models have showcased remarkable capabilities in generating realistic and diverse images in response to textual prompts. However, despite the impressive results, diffusion models often struggle to accurately follow complex prompts that require specific capabilities to understand. Fig. 1 shows that Stable Diffusion (Rombach et al., 2022), even the latest SDXL (Podell et al., 2023), often could not generate a certain number of objects or understand negation in the prompt. It also struggles with spatial reasoning or associating attributes correctly with objects.

One potential solution to address this issue is of course to gather a comprehensive multi-modal dataset comprising intricate captions and train a text-to-image diffusion model for enhanced prompt understanding.

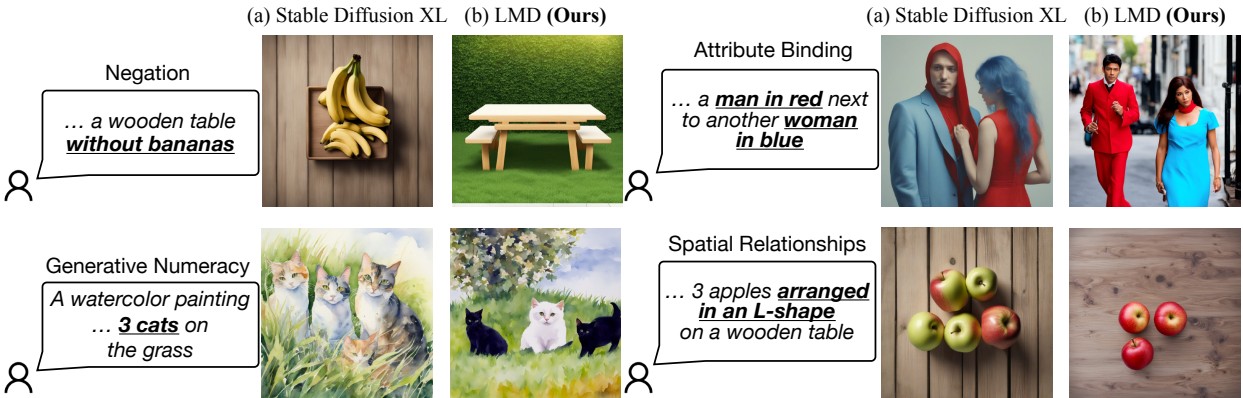

Figure 1: **(a)** Text-to-image diffusion models such as SDXL (Podell et al., 2023) often struggles to accurately follow prompts that involve negation, numeracy, attribute binding, or spatial relationships. **(b)** Our method LMD achieves enhanced prompt understanding capabilities and accurately follows these types of prompts.

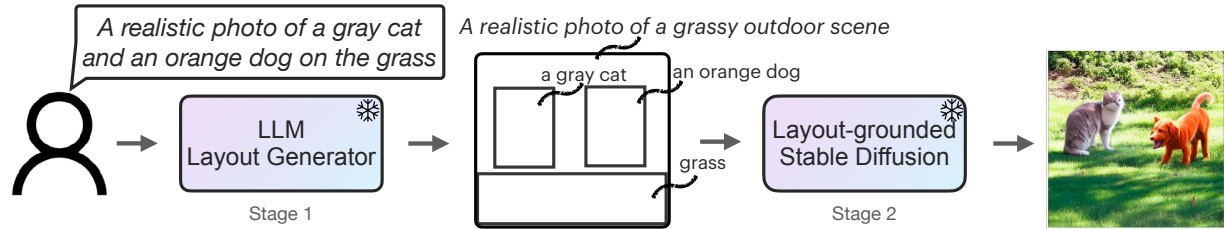

Figure 2: **Our proposed LMD enhances prompt understanding in text-to-image diffusion models through a novel two-stage generation process**: **1)** An LLM layout generator takes a prompt from the user and outputs an image layout in the form of captioned bounding boxes. **2)** A stable diffusion model guided by our layout-grounded controller generates the final image. Both stages utilize frozen pretrained models, which makes our method applicable to off-the-shelf LLMs and other diffusion models without grounding in their training objectives.

Nonetheless, this approach presents notable drawbacks. It requires considerable time and resources to curate a diverse and high-quality multi-modal dataset, not to mention the challenges associated with training or fine-tuning a diffusion model on such extensive data.

In contrast, we propose a novel *training-free* method that equips the diffusion model with an LLM that provides grounding for enhanced prompt understanding. Our method **LLM**-grounded **D**iffusion (LMD) consists of a two-stage generation process as shown in Fig. 2.

In the first stage of our method, we adapt an LLM to be a text-grounded layout generator through in-context learning. Given a prompt describing the desired image, the LLM generates scene layouts in the form of captioned bounding boxes, with a background caption and a negative prompt for what to avoid in generation.

In the second stage, we introduce a novel controller that guides an existing diffusion model without grounding in its training objective (e.g., Stable Diffusion) to follow the layout grounding generated in the first stage. In contrast to previous and concurrent works on region control (e.g., Bar-Tal et al. (2023); Chen et al. (2023); Xie et al. (2023)) that apply *semantic* control to certain spatial regions, our approach allows precise control over object *instances* in designated regions.

Notably, both stages utilize frozen pretrained models *off-the-shelf*, making our method applicable to LLMs and diffusion models trained independently *without any LLM or diffusion model parameter optimization*.

In addition to enhanced prompt understanding, our method also naturally enables instruction-based scene specification with multiple rounds of user requests (Fig. 3) and image generation from prompts in languages not supported by the base diffusion model (Fig. I.1) without additional training.

Shown in Fig. 1, LMD provides a unified solution to several caveats in prompt understanding *at once* and enables accurate and high-quality image generation from complex prompts. We demonstrate that a diffusion

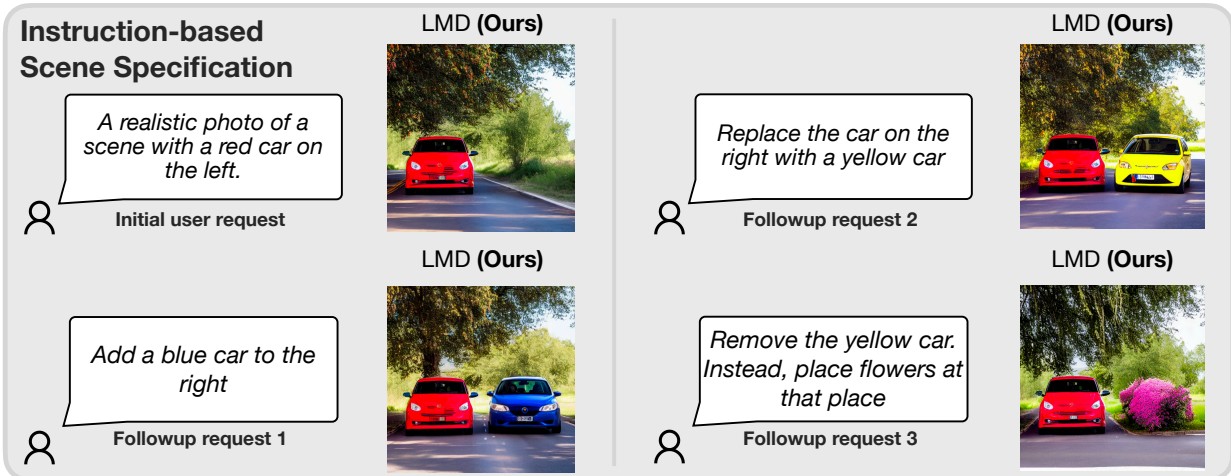

Figure 3: **LMD naturally enables instruction-based multi-round scene specification** and is able to adapt subsequent rounds of generation according to users' followup instructions and clarifications.

model grounded with LLM-generated layouts outperforms its base diffusion model and several recent baselines, *doubling* the average generation accuracy across four tasks. **Our primary contributions include:**

1. We propose a training-free two-stage generation pipeline that introduces LLMs to improve the prompt understanding ability of text-to-image diffusion models.

2. We introduce layout-grounded Stable Diffusion, a novel controller that steers an off-the-shelf diffusion model to generate images grounded on instance-level box layouts from the LLM.

3. LMD enables instruction-based scene specification and allows broader language support in the prompts.

4. We propose a benchmark to assess the prompt understanding ability of a text-to-image model and demonstrate the superior performance of LMD over recent baselines.

We expect LMD to empower users with more precise control of text-to-image diffusion models. Our code, demo, and benchmark are publicly available.

## 2   Related Work

**Text-to-image diffusion models.** High-quality image generation from textual descriptions with diffusion models has been popular recently (Ramesh et al., 2022; Saharia et al., 2022; Rombach et al., 2022; Podell et al., 2023). Despite the impressive visual quality, these models still tend to exhibit unsatisfactory performance when it comes to complex prompts that involve skills such as binding attributes to objects and spatial reasoning (Ramesh et al., 2022).

**LLMs for visual grounding.** Many multi-modal models benefit from integrating LLMs for grounding vision models. BLIP-2 (Li et al., 2023a) bootstraps vision-language pre-training from a frozen image encoder and an LLM. Flamingo (Alayrac et al., 2022) tackles tasks such as few-shot visual question-answering and captioning tasks. Gupta et al. (2021) uses Transformer (Vaswani et al., 2017) for layout prediction but focuses on generating layouts for a limited closed set of object classes in the annotated training set and thus is not able to generate layouts for objects not in the training set. Wu et al. (2023) and Koh et al. (2023) also involve LLMs in conditional image generation. However, these methods still rely on CLIP text embeddings to convey the information to the diffusion model. Therefore, they often exhibit insufficient control compared to our method, which explicitly asks the LLM to reason about the spatial composition of different objects and poses direct spatial control. Concurrent to our work, LayoutGPT (Feng et al., 2023) proposes prompting an LLM for layout generation in a CSS structure. While LayoutGPT depends on a dataset annotated with boxes and captions to retrieve relevant in-context examples for the LLM, our method demonstrates that the ability for

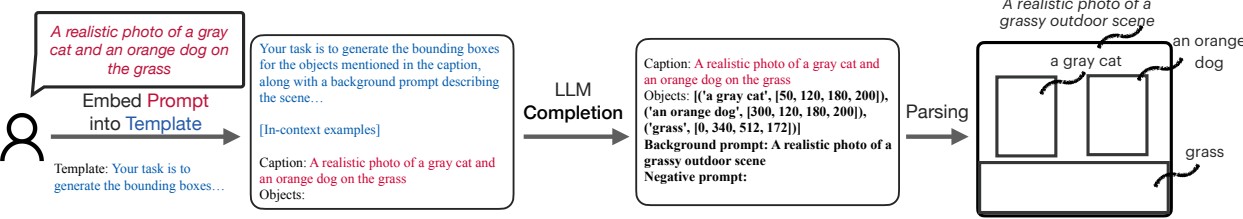

Figure 4: **In stage 1, LMD generates an image layout from a user prompt.** LMD embeds the user prompt into a template with instructions and in-context examples. An LLM is then queried for completion. Finally, the LLM completion is parsed to obtain a set of captioned bounding boxes, a background caption, and an optional negative prompt.

generating high-quality layouts is already present in pretrained LLM weights and can be prompted with a fixed set of in-context examples without external annotations.

**Spatially-conditioned image generation methods.** These methods create images based on given priors such as poses, segmentation maps, strokes, and layouts. Prior to the popularity of diffusion models, SPADE (Park et al., 2019), BlobGAN (Epstein et al., 2022), and Layout2Im (Zhao et al., 2019) synthesize photorealistic images from a given layout. Xu et al. (2017); Johnson et al. (2018); Herzig et al. (2020) generate images with scene graphs. ControlNet (Zhang & Agrawala, 2023), SpaText (Avrahami et al., 2023), LayoutDiffuse (Cheng et al., 2023), LayoutDiffusion, (Zheng et al., 2023), GLIGEN (Li et al., 2023b) and ReCo (Yang et al., 2023) propose training-based adaptation on the diffusion models for spatially-conditioned image generation, with Li et al. (2023b) and Yang et al. (2023) supporting open-vocabulary labels for layout boxes. However, these methods *rely on annotated external datasets* such as COCO (Lin et al., 2014) to supply images with annotations such as boxes and captions. Furthermore, training-based adaptation makes the model incompatible to add-ons such as LoRA weights (Hu et al., 2021) and renders it difficult to train a new LoRA model from a training set without box annotations. In contrast, we propose a training-free generation controller that steers *existing* text-to-image diffusion models that are *not specifically trained* for layout-grounded image generation and does *not* require external datasets. Furthermore, our method can also integrate with training-based methods for further improvements.

Very recently, Bar-Tal et al. (2023); Chen et al. (2023); Xie et al. (2023) allow training-free region control in image generation and share a similar task formulation to our layout-to-image stage. However, these works ground the image generation on the region *semantics* and pose little control over the number of object instances inside each semantic region, whereas our method focuses on grounding generation on *instances*.

Similar to our instruction-based scene specification, Brooks et al. (2023) recently proposed instruction-based image editing. Wu et al. (2023) and Gupta & Kembhavi (2023) also allow using external image editing models in an LLM-driven dialog. Different from these methods, we aim to edit the *scene layout* rather than the *image pixels*, which easily allows support for a greater set of instructions such as swapping/moving objects.

## 3 LLM-grounded Diffusion

In this section, we introduce our method **LLM**-grounded **D**iffusion (LMD). LMD focuses on the text-to-image generation setting, which involves generating image $\mathbf{x}_0$ given text prompt $\mathbf{y}$. Our method generates an image in two stages: text-grounded layout generation (Section 3.1) and layout-grounded image generation (Section 3.2). The layout-to-image stage of our method LMD builds upon the latent diffusion framework (Rombach et al., 2022), for which we refer readers to Appendix A for preliminaries.

### 3.1 LLM-based Layout Generation

To generate the layout of an image, our method embeds the input text prompt $\mathbf{y}$ into a template and queries an LLM for completion (Fig. 4).

**Layout representation.** LMD's layout representation comprises two components: **1)** a captioned bounding box for each foreground object, with coordinates specified in the *(x, y, width, height)* format, and **2)** a simple

and concise caption describing the image background along with an optional negative prompt indicating what should not appear in a generated image. The negative prompt is an empty string when the layout does not impose restrictions on what should not appear.

**Instructions.** Our text instructions to the LLM consist of two parts:
1. Task specification:

> *Your task is to generate the bounding boxes for the objects mentioned in the caption, along with a background prompt describing the scene.*

2. Supporting details:

> *The images are of size 512×512... Each bounding box should be in the format of ... If needed, you can make reasonable guesses.*

**In-context learning.** Similar to Brooks et al. (2023), we provide the LLM with manually curated examples after the task description. Through these examples, we clarify the layout representation and provide preferences to disperse ambiguity. An example is shown as follows:

> *Caption: A watercolor painting of a wooden table in the living room with an apple on it*
> *Objects: [('a wooden table', [65, 243, 344, 206]), ('an apple', [206, 306, 81, 69])]*
> *Background prompt: A watercolor painting of a living room*
> *Negative prompt:*

To ensure precise layout control, we adhere to two key principles in our example design: **1)** Each object instance is represented by a single bounding box. For instance, if the prompt mentions four apples, we include four boxes with "an apple" in each caption. **2)** We leave no foreground objects specified in the boxes to the background caption to ensure all foreground objects are controlled by our layout-grounded image generator (Section 3.2). These principles allow for accurate and instance-controlled layout generation.

**LLM completion.** After providing the in-context examples, we query the LLM for completion:

> *Caption: [input prompt from the user]*
> *Objects: [start of LLM completion]*

The resulting layout from the LLM completion is then parsed and used for the subsequent image generation process. We refer readers to the Appendix K for our complete prompt.

## 3.2 Layout-grounded Stable Diffusion

In this stage, we introduce a controller to ground the image generation on the LLM-generated layout. While previous training-free region control methods (Bar-Tal et al., 2023; Chen et al., 2023; Xie et al., 2023) apply *semantic* guidance through regional denoising or attention manipulation, these methods lack the ability to control the number of objects within a semantic region. This limitation arises as the different instances are often indistinguishable in either the latent space or the attention map, hindering instance-level control.

In contrast, LMD enables *instance*-level grounding by first generating masked latents for each individual bounding box and then composing the masked latents as priors to guide the overall image generation. This allows for precise placement and attribute binding for each object instance.

**Per-box masked latents.** While diffusion models lack inherent instance-level distinction in their latent space or attention maps for fine-grained control, we observe that they are often able to generate images with one specified instance. Hence, we process one foreground box at a time for instance-level grounding.

As depicted in Fig. 5(a), for each foreground object $i$, we first generate an image with a single instance by denoising from $\mathbf{z}_T^{(i)}$ to $\mathbf{z}_0^{(i)}$, where $\mathbf{z}_t^{(i)}$ refers to the latents of object $i$ at denoising timestep $t$.[1] In this denoising process, we use *"[background prompt] with [box caption]"* (e.g., *"a realistic image of an indoor scene with a gray cat"*) as the text prompt for denoising. The initial noise latent is shared for all boxes to ensure globally coherent viewpoint, style, and lighting (i.e., $\mathbf{z}_T^{(i)} = \mathbf{z}_T, \forall\, i$).

---

[1] We refer readers to Appendix A for definitions of the terms, such as latents, that are introduced in the latent diffusion framework.

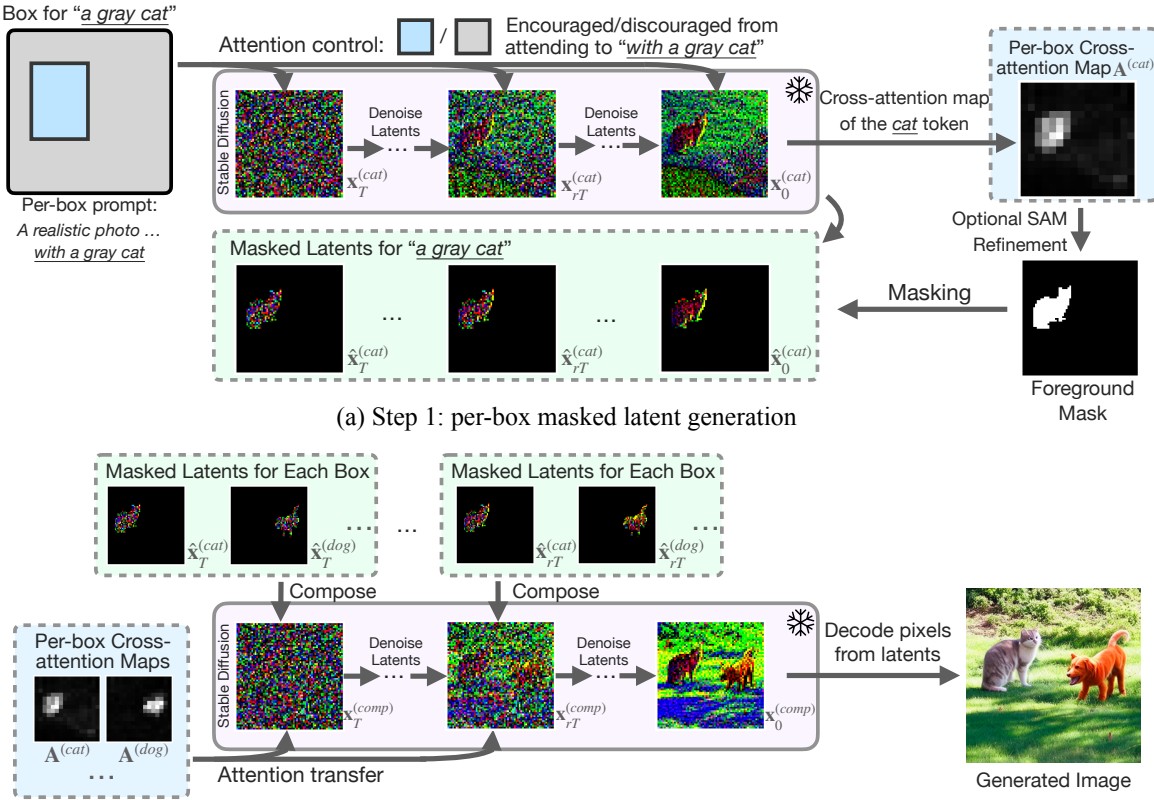

(a) Step 1: per-box masked latent generation

(b) Step 2: overall image generation with masked latents as priors

Figure 5: **In stage 2, we introduce a novel layout-grounded controller that guides stable diffusion to generate images based on the layout obtained from the previous stage.** Our layout-grounded image generation process consists of two steps: **(a)** generating masked latents for each box specified in the layout, with attention control ensuring that the object is placed in the designated box; and **(b)** composing the masked latents as priors to guide the image generation to adhere to the specified layout.

To ensure the object aligns with the bounding box, we manipulate the cross-attention maps $\mathbf{A}^{(i)}$ of the noise-prediction network.[2] Each map describes the affinity from pixels to text tokens:

$$\mathbf{A}^{(i)}_{uv} = \texttt{Softmax}(\mathbf{q}_u^T \mathbf{k}_v) \tag{1}$$

where $\mathbf{q}_u$ and $\mathbf{k}_v$ are linearly transformed image feature at spatial location $u$ and text feature at token index $v$ in the prompt, respectively.

Following Chen et al. (2023); Xie et al. (2023), we strengthen the cross-attention from pixels inside the box to tokens associated with the box caption while attenuating the cross-attention from pixels outside the box. To achieve this, we define a simple energy function:

$$E(\mathbf{A}^{(i)}, i, v) = -\texttt{Topk}_u(\mathbf{A}_{uv} \cdot \mathbf{b}^{(i)}) + \omega\texttt{Topk}_u(\mathbf{A}_{uv} \cdot (1 - \mathbf{b}^{(i)})) \tag{2}$$

where $\cdot$ is element-wise multiplication, $\mathbf{b}^{(i)}$ is a rectangular binary mask of the box $i$ with the region in the box set to 1, $\texttt{Topk}_u$ takes the average of top-k values across the spatial dimension $u$, and $\omega = 4.0$. The energy function is minimized by updating the latent before each denoising step:

$$\mathbf{z}_t^{(i)} \leftarrow \mathbf{z}_t^{(i)} - \eta\nabla_{\mathbf{z}_t^{(i)}} \sum_{v \in V_i} E(\mathbf{A}^{(i)}, i, v) \tag{3}$$

$$\mathbf{z}_{t-1}^{(i)} \leftarrow \texttt{Denoise}(\mathbf{z}_t^{(i)}) \tag{4}$$

where $\eta$ is the guidance strength; the set $V_i$ contains the token indices for the box caption in the prompt for box $i$ (e.g., while generating the masked latents for a box $i$ with caption *"a gray cat"*, $V_i$ indicates the indices of tokens that correspond to the box caption in the per-box denoising text prompt *"[background prompt] with a gray cat"*). $\texttt{Denoise}(\cdot)$ denotes one denoising step in the latent diffusion framework.

---

[2]The cross-attention layer index is omitted for simplicity. We sum the energy values for all selected layers during optimization.

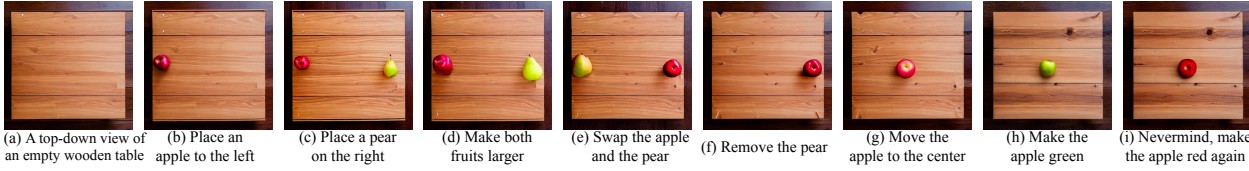

(a) A top-down view of an empty wooden table · (b) Place an apple to the left · (c) Place a pear on the right · (d) Make both fruits larger · (e) Swap the apple and the pear · (f) Remove the pear · (g) Move the apple to the center · (h) Make the apple green · (i) Nevermind, make the apple red again

Figure 6: **LMD and LMD+ support instruction-based scene specification, empowering the users to add/move/remove objects, modify object attributes, and clarify the prompt in multiple rounds of dialog. (a)**: the initial prompt for the scene; **(b)-(i)**: eight subsequent instructions that *sequentially* modify the scene. By separating the generation of each foreground object as well as the background, LMD ensures consistent image generation when the same seed is used for image generation throughout the dialog.

After generation, we obtain the cross-attention map that corresponds to the box caption, which serves as a saliency mask for the object. We optionally use SAM (Kirillov et al., 2023) to refine the quality of the mask. This can be done by querying either with the pixel location that has the highest saliency or with the layout box. The functionality of SAM can also be replaced by a simple thresholding, as experimented in Section 4.3. With the refined mask for exactly one foreground instance, denoted as $\mathbf{m}^{(i)}$, we perform element-wise multiplication between the mask and the latent at each denoising step to create a sequence of *masked* instance latents $(\hat{\mathbf{z}}_t^{(i)})_{t=0}^T$:

$$\hat{\mathbf{z}}_t^{(i)} = \mathbf{z}_t^{(i)} \otimes \mathbf{m}^{(i)} \tag{5}$$

**Masked latents as priors for instance-level control.** The masked instance latents $(\hat{\mathbf{z}}_t^{(i)})_{t=0}^T$ are then leveraged to provide instance-level hints to the diffusion model for the overall image generation. As illustrated in Fig. 5(b), during each denoising time step in the early denoising process, we place each masked foreground latents $\hat{\mathbf{z}}_t^{(i)}$ onto the composed latents $\mathbf{z}_t^{(\text{comp})}$:

$$\mathbf{z}_t^{(\text{comp})} \leftarrow \texttt{LatentCompose}(\mathbf{z}_t^{(\text{comp})}, \hat{\mathbf{z}}_t^{(i)}, \mathbf{m}^{(i)}) \quad \forall i \tag{6}$$

where $\mathbf{z}_T^{(\text{comp})}$ is initialized from $\mathbf{z}_T$ for foreground generation for consistency, and $\texttt{LatentCompose}(\mathbf{z}_t^{(\text{comp})}, \hat{\mathbf{z}}_t^{(i)}, \mathbf{m}^{(i)})$ simply puts the masked foreground latents $\hat{\mathbf{z}}_t^{(i)}$ onto the corresponding location on $\mathbf{z}_t^{(\text{comp})}$.

Since diffusion models tend to generate the object placement in the initial denoising steps and then object details in later steps (Bar-Tal et al., 2023), we only compose the latents from timestep $T$ to $rT^3$, where $r \in [0, 1]$ balances instance control and image coherency. By primarily intervening during the steps for object placement, our method merely provides instance-level layout hints rather than forcing each masked region of the resulting generation to look the same as the per-box generation.

To make our guidance more robust, we further transfer the cross-attention maps from per-box generation to the corresponding regions in the composed generation by adapting the energy function:

$$E^{(\text{comp})}(\mathbf{A}^{(\text{comp})}, \mathbf{A}^{(i)}, i, v) = E(\mathbf{A}^{(\text{comp})}, i, v) + \lambda \sum_{u \in V_i'} \left| \mathbf{A}_{uv}^{(\text{comp})} - \mathbf{A}_{uv}^{(i)} \right| \tag{7}$$

where $\lambda = 2.0$ and the energy value of each box $i$ is summed up for optimization. $V_i'$ denotes the indices of tokens that correspond to the box caption in the text prompt for the overall denoising process, similar to the definition of $V_i$ in Eq. (3).

In this way, our controller conditions the diffusion model to generate one instance at each masked location, with the final generation natural and coherent in terms of foreground-background composition.

Finally, we decode latents $\mathbf{z}_0^{(\text{comp})}$ to pixels $\mathbf{x}_0$ via the diffusion image decoder. We refer readers to Appendix B for the overall pseudo-code for layout grounding.

**Integration with training-based methods.** Our training-free controller can also be applied along with training-based methods such as GLIGEN (Li et al., 2023b) to leverage instance-annotated external datasets when available. Since GLIGEN trains adapter layers taking box inputs, the integration with GLIGEN, denoted as LMD+, involves adopting its adapter weights and passing the layout guidance to the adapter

---

[3]In the notation of this work, the denoising process starts from time step $T$ to step 0.

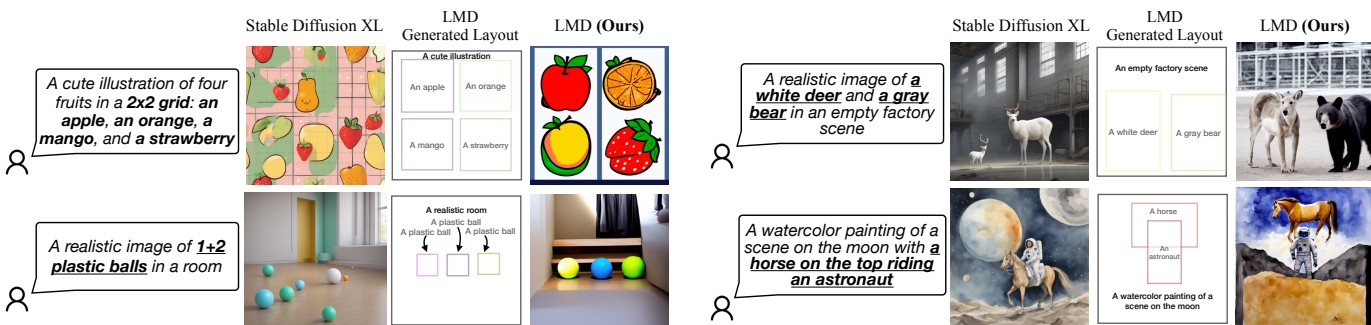

Figure 7: LMD outperforms its base text-to-image diffusion model Podell et al. (2023) in accurately following the prompts that require spatial and language reasoning. Best viewed when zoomed in.

layers. Note that LMD+ uses adapters along with the instance-level guidance introduced above, which greatly surpasses only using GLIGEN adapters, as shown in Table 2. We achieve further enhanced instance and attribute control without additional training through this integration.

### 3.3 Additional Capabilities of LMD

Our LLM-grounded generation pipeline allows for two additional capabilities without additional training.

**Instruction-based scene specification.** Leveraging an LLM that supports multi-round dialog (e.g., GPT-3.5/4), LMD empowers the users to specify the desired image with multiple instructions following an initial prompt (Fig. 3). Specifically, after the initial image generation, a user can simply give clarifications or additional requests to the LLM. With the updated layout from the LLM, we can leverage LMD again to generate images with the updated layout. Updating the layout rather than the raw image gives LMD several advantages, as demonstrated in Fig. 6: **1)** Our generation remains consistent after multiple rounds of requests instead of gradually drifting away from the intial image. **2)** LMD can handle requests that involve spatial reasoning, which are the limitations of previous instruction-based image editing method Brooks et al. (2023). In contrast, we demonstrate that VisualChatGPT Wu et al. (2023), which equips ChatGPT with tools such as Brooks et al. (2023), is not able to follow the instructions in Fig. 6, especially for spatial instructions over multiple iterations of dialog. We refer interested readers to Appendix G for the comparison. This capability applies to both LMD and LMD+. We also show additional use cases in Fig. C.1 in Appendix C. Our LMD can handle requests for open-ended scene adjustments, offer suggestions for the current scene, understand user requests within the dialog context, and allow the users to try out different detailed adjustments while preserving the overall image style and layout, facilitating fine-grained content creation.

**Supporting more languages.** By giving an in-content example of a non-English user prompt and an English layout output[4], the LLM layout generator accepts non-English user prompts and outputs layouts with English captions. This allows generation from prompts in languages not supported by the underlying diffusion model *without additional training* (Fig. I.1). We refer readers to Appendix I for additional details.

## 4 Evaluation

### 4.1 Qualitative Comparison

**Setup.** We qualitatively compare our approach with Stable Diffusion (SD, Rombach et al. (2022); Podell et al. (2023)). SD family is also chosen as our underlying base model for layout-grounded image generation given its strong capabilities and widespread adoption in text-to-image generation research. Thanks to the training-free nature of our work, our method is applicable to various diffusion models without additional training. Therefore, for Fig. 1, 7, and 9, we use the largest Stable Diffusion model SDXL as the base model of LMD and compare against SDXL as a baseline (see Appendix H for details). For all other settings, we

---

[4]We simply translate the example prompt input in our last example to the desired language, while keeping its layout in English.

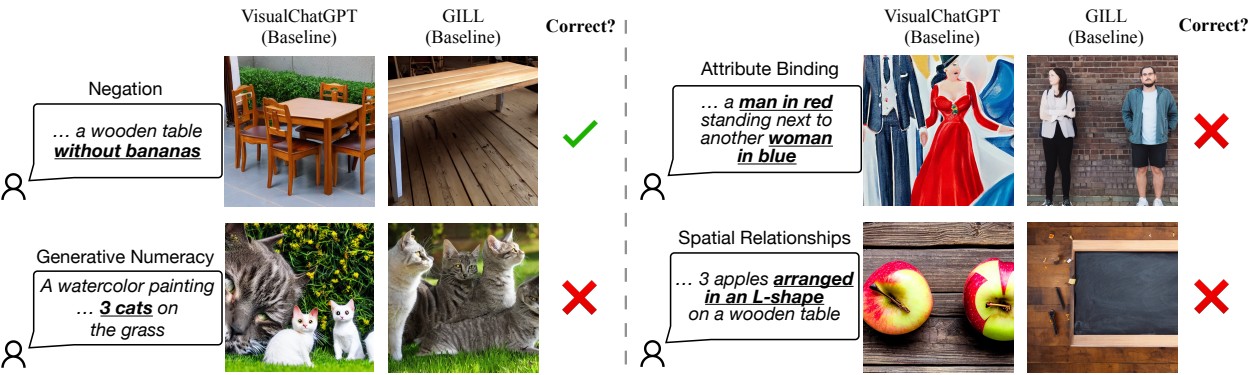

Figure 8: We qualitatively compare with VisualChatGPT (Wu et al., 2023) and GILL (Koh et al., 2023) that also leverage LLMs in the image generation pipelines. **Both baselines lack the ability to accurately follow the prompts for three out of four tasks that our method can solve in Fig. 1 and F.1.**

| Tasks | SD | Accuracy | |
| --- | --- | --- | --- |
| | | **LMD** | **LMD+** |
| Negation | 28% | **100% (3.6×)** | **100% (3.6×)** |
| Generative Numeracy | 39% | **62% (1.6×)** | **86% (2.2×)** |
| Attribute Binding | 52% | **65% (1.3×)** | **69% (1.3×)** |
| Spatial Relationships | 28% | **79% (2.8×)** | **67% (2.4×)** |
| Average | 37% | **77% (2.1×)** | **81% (2.2×)** |

Table 1: With guidance from the LLM-based layout generator and our novel layout-grounded controller, **our LMD significantly outperforms the Stable Diffusion model (SD) that we use under the hood in four tasks benchmarking prompt-following abilities.** LMD denotes our method directly applied on SD. LMD+ denotes additionally integrating pretrained GLIGEN (Li et al., 2023b) adapters into our controller.

use Stable Diffusion v1.5 as the base model unless stated otherwise. We use `gpt-4` (OpenAI, 2023) for layout generation for all qualitative comparisons. **Results.** In Fig. 1 and 7, we observe that our two-stage text-to-image generation approach greatly enhances prompt following ability compared to our base model by generating images that align with the layouts from the LLM.

**Comparing with other LLM-based image generators.** VisualChatGPT (Wu et al., 2023) and GILL (Koh et al., 2023) also leverage LLMs as a part of the image generation pipelines. Both works leverage SD as the underlying image generation model. VisualChatGPT treats SD as a module that can be used by the LLM and passes text caption to it, and GILL outputs a embedding in place of the text embedding for SD. Since both methods utilize LLMs to only provide conditions to SD in the form of text embeddings, these methods still inherit the problems of insufficient control of text embeddings from the base SD model. In contrast, our method asks the LLM to explicitly reason about the spatial relationships and applies direct spatial control on our underlying diffusion model, thereby bypassing the bottleneck of the text embedding representation that does not accurately convey spatial information. As shown in Fig. 8, neither method accurately follows text prompts of several categories that our method is able to correctly generate in Fig. 1 and Fig. F.1 in Appendix F. Furthermore, although the involvement of LLM in VisualChatGPT and GILL also potentially allows multi-round instruction-based scene specification (Section 3.3), we empirically observe that the generated images quickly deviate from the scene of "a wooden table" starting from the second iteration in Fig. G.1 in Appendix G, with the final generation being incomprehensible.

| | Accuracy | | | | |
|---|---|---|---|---|---|
| Stage 1/Stage 2 | Negation | Numeracy | Attribute | Spatial | Average |
| *Training-free methods:* | | | | | |
| LMD/MultiDiffusion (Bar-Tal et al., 2023) | **100%** | 30% | 42% | 36% | 52.0% |
| LMD/Backward Guidance (Chen et al., 2023) | **100%** | 42% | 36% | 61% | 59.8% |
| LMD/BoxDiff (Xie et al., 2023) | **100%** | 32% | 55% | 62% | 62.3% |
| **LMD/LMD (Ours)** | **100%** | **62%** | **65%** | **79%** | **76.5%** (+ 14.2) |
| *Training-based methods:* | | | | | |
| LMD/GLIGEN (Li et al., 2023b) | **100%** | 57% | 57% | 45% | 64.8% |
| **LMD/LMD+ (Ours)** | **100%** | **86%** | **69%** | **67%** | **80.5%** (+ 15.7) |
| **LMD/LMD+ (Ours, GPT-4)** | **100%** | **84%** | **79%** | **82%** | **86.3%** (+ 21.5) |
| *Evaluating generated layouts only (upper bound for image generation):* | | | | | |
| LMD/- | 100% | 97% | 100% | 99% | 99.0% |

Table 2: **Ablations on layout-to-image methods as stage 2 with our LLM layout generator as stage 1. Our proposed layout-grounded controller performs the best among them.** Our controller could also be applied on top of training-based GLIGEN (Li et al., 2023b), denoted as LMD+, for additional improvements. Finally, the LLM-generated layouts almost always align with the prompt, highlighting that the bottleneck is the layout-grounded image generation. The scores for negation task are high because we pass the negative prompts generated by the LLM to the underlying diffusion model, which does not depend on the stage 2 implementation.

## 4.2 Quantitative evaluation

### 4.2.1 Proposed benchmark

We propose a text-to-image evaluation benchmark that includes four tasks: negation, generative numeracy, attribute binding, and spatial reasoning. Negation and generative numeracy involve generating a specific number of objects or not generating specific objects. Attribute binding involves assigning the right attribute to the right object with multiple objects in the prompt. Spatial reasoning involves understanding words that describe the relative locations of objects. For each task, we programmatically compose 100 prompts and query each model for text-to-image generation, with 400 prompts in total. `gpt-3.5-turbo` (Brown et al., 2020) is used in LMD for the benchmarks. We also implemented LMD+, a LMD variant that integrate pretrained GLIGEN (Li et al., 2023b) adapters into our controller without further training. We refer readers to Appendix J for details.

**Detection-based evaluation.** We use an open-vocabulary object detector, OWL-ViT (Minderer et al., 2022), to obtain bounding boxes for the objects of interest. We then check whether each generated image satisfies the requirements in the prompt. The accuracy of each task is computed by calculating the proportion of the image generations that match their corresponding prompts over all generations.

**Results.** As presented in Table 1, our model shows significant improvements in generation accuracy, ranging from $1.3\times$ to $3.6\times$ compared to SD across four tasks and *doubling* the accuracy on average. Notably, LMD achieves image generation accuracy that is *more than twice* of the SD accuracy for the spatial relationships and the negation task. This highlights the utility of the grounding image generation on the LLM layout generator. Furthermore, when additionally integrating GLIGEN to our pipeline to leverage in-domain instance-annotated data, our method, denoted as LMD+, achieves additional improvements.

## 4.3 Ablation Study

**Layout-to-image stage.** *Comparing with other layout-to-image methods.* As shown in Table 2, compared with training-free layout-to-image generation methods that perform *semantic*-level grounding, our proposed layout-grounded controller provides much better *instance*-level grounding. This is justified by the

| Method | Image Accuracy Average of 4 tasks |
|---|---|
| SD v1.5 (Default) | 37% |
| LMD (on SDv1.5) (**Ours**, default) | **77%** **(2.1×)** |
| SD v2.1 | 38% |
| LMD (on SDv2.1) (**Ours**) | **77%** **(2.0×)** |

Table 3: **LMD achieves comparable gains when adapted to Stable Diffusion v2.1 without any hyperparameter tuning or model training.** This shows a promising signal that the gains from our method could carry along with the enhancement of diffusion models. The performance of our method could potentially be improved further with additional hyperparameter tuning.

| Method | Image Accuracy Average of 4 tasks |
|---|---|
| LMD (w/o SAM) | 72.8% |
| LMD (with SAM) | **76.5%** |
| LMD+ (w/o SAM) | **82.8%** |
| LMD+ (with SAM) | 80.5% |

Table 4: **Ablations on using SAM vs using simple attention thresholding in stage 2.** While removing SAM leads to a slight degradation in LMD, removing SAM leads to even better performance in LMD+.

fact that our training-free controller even *surpasses training-based method* GLIGEN (Li et al., 2023b) in the generative numeracy task, despite not trained with any instance-level annotation. Furthermore, our controller also sigficantly surpasses training-based method GLIGEN (Li et al., 2023b) in attribute binding and spatial reasoning task. When integrated with GLIGEN to leverage instance-annotated datasets, our integration, denoted as LMD+, allows for further improvements without the need for additional training. ***Switching the base diffusion model without hyperparameter tuning.*** As shown in Table 3, thanks to our training-free nature, LMD maintains the gains to the base model (around $2\times$ performance boost) when we switch the base diffusion model from SDv1.5 to SDv2.1 *without tuning any hyperparameters*, including $\lambda$ and $\omega$ that are introduced by our method.[5] This showcases the potential of integrating LMD with future diffusion models. ***Using SAM vs a simple attention threshold to obtain the per-box mask.*** Instead of using SAM to obtain the mask for each box, we also explored an approach that does not require an additional segmentation module. Alternatively, we sort the pixels in each box according to their attention value with respect to the box caption and pick the top 75% pixels in each box with the highest attention as the mask for the box. As shown in Table 4, the impact of SAM is different for LMD/LMD+. In LMD, since the attention-based guidance is less spatially accurate with respect to the layout boxes, SAM helps to obtain the right mask that covers the object. Therefore, removing SAM leads to a slight degradation in LMD. In LMD+, since the guidance is more spatially accurate, SAM is no longer necessary most of the time. Instead, SAM sometimes picks a region that includes the background, causing confusion and reduced performance. Therefore, removing SAM slightly improves the results in LMD+. We make SAM an optional choice (as described in Fig. 2) but still recommend it for LMD and enable it by default. We refer readers to Appendix D for additional ablations on the values of the hyperparameters.

**Text-to-layout stage.** *Ablating in-context examples.* In addition to using the seven fixed in-context examples provided in Table K.2 by default, we also vary the number of in-context examples given to the LLM (i.e., "shots"). We show in Table 5 that while GPT-3.5 benefits from more in-context examples, GPT-4 is able to successfully generate all the layouts even when given only one in-context example. Note that we also observe GPT-4 to still be able to generate layouts *without any in-context examples* (i.e., given only the text instructions). However, since no examples are offered as references in this zero-shot setting, the format of LLM outputs are observed to differ in different runs, making it hard to parse with a program. Since it is much easier to convey the format through an example than through language instructions, we recommend having at least one example. Our observation shows that LLMs *already learn the ability to generate object boxes during pretraining* and do not need us to convey through many in-context examples. ***Varying the model types and the sizes of the LLMs.*** We also ablate the LLMs used for text-to-layout generation, including using self-hosted LLMs with public weights (Mahan et al., 2023; Touvron et al., 2023; Mukherjee et al., 2023; Jiang et al., 2024). The results show that the capability to generate high-quality layouts are *not limited to proprietary LLMs*, and larger LLMs offer much better layout generation capabilities. We refer the readers to Appendix D and Appendix E for more ablations and investigations.

---

[5]The hyperparameters inherited from the latent diffusion framework, such as the number of denoising steps, are also unchanged.

|  | Layout Accuracy (4 tasks) | |
|---|---|---|
| # shots | `gpt-3.5-turbo` | `gpt-4` |
| 1 Shot | 89.8% | **100.0%** |
| 4 Shots | 96.3% | **100.0%** |
| 7 Shots | **99.0%** | **100.0%** |

Table 5: **Ablations on the number of in-context examples ("shots") given to the LLM.** While GPT-3.5 benefits from more in-context examples, GPT-4 already excels in layout generation even with only one example.

|  | Color | Shape | Texture | Spatial |
|---|---|---|---|---|
| SDv1 | 0.3765 | 0.3576 | 0.4156 | 0.1246 |
| LMD (on SDv1) | **0.5495** | **0.5462** | **0.5241** | **0.2570** |
| SDv2 | 0.5065 | 0.4221 | 0.4922 | 0.1342 |
| LMD (on SDv2) | **0.5736** | **0.5334** | **0.5227** | **0.2704** |

Table 6: **Our method surpasses the base diffusion models SDv1 and SDv2 on T2I-CompBench (Huang et al., 2023) on all four tasks without additional training.**

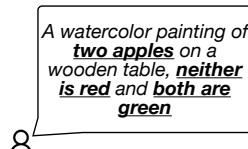 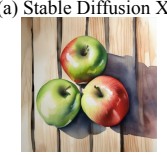 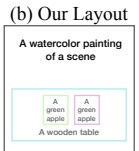 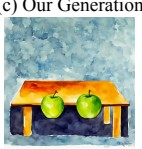 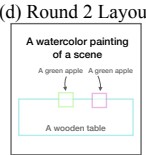 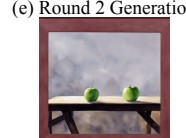

Figure 9: **A failure case** occurs when our method, shown in **(c)**, generates objects in unintentional viewpoints and sizes due to the ambiguity in the generated layout. The LLM-generated layout **(b)** is suitable for close-up top-down view of a small table, but the layout-to-image model assumes a side view and thus fails to generate a feasible image. Nevertheless, our method still provides more interpretability through the intermediate layout **(b)** compared to baseline SDXL **(a)**. With an additional request for the side view and correct object sizes, the LLM adjusted the layout in **(d)** and the final generation **(e)** is aligned with the text prompt.

## 4.4 T2I-CompBench

In addition to our proposed benchmark with detection-based evaluation, we evaluate our method on T2I-CompBench (Huang et al., 2023) that additionally uses visual question answering (VQA) models for generation evaluation. The color, shape, and texture tasks employ BLIP (Li et al., 2022) in a VQA setting, while the spatial task uses UniDet (Zhou et al., 2022) for evaluation. As shown in Table 6, our method LMD, when applied on either SDv1 or SDv2, improves the performance on all four tasks. Additional ablations are in Table D.4.

## 4.5 Evaluator-based Assessment

**Setting.** We also assess the prompt following ability of our method and vanilla SD, the base diffusion model that our method uses under the hood. We randomly selected 10 text prompts from our proposed benchmark and generated a pair of images per text prompt, one with our LMD+ and one with the base model SD.[6] We then invited 11 evaluators to compare each image pair and answer two questions:

1. Question 1: Which image aligns better with the text prompt?
2. Question 2: Which image has a more natural and coherent foreground-background composition?

In addition to an option for preferring each image, a "similar" option is also provided for each pair.

**Results.** We average the scores across 110 responses. The results show that our method LMD+ got **88.18%** (vs 10.90% for SD) for the first question and **35.45%** (vs 31.81% for SD) for the second question. This indicates that our method generates images that accurately align with the prompt compared to the baseline SD without degradation of naturalness or coherency.

---

[6]The base model of LMD+ is technically GLIGEN, which itself is built on SD. However, in our setting where only text is used as input, the conditioning scheme of GLIGEN simply becomes degenerate. Therefore, we compare with SD directly instead.

## 5    Discussions

Since we use models off-the-shelf, the LLM may generate layouts that are ambiguous to the diffusion model. For example, the layout in Fig. 9(b) is feasible for a top-down close-up image, but the diffusion model generates an image viewing from the side. This makes the apples not on the table in Fig. 9(c). Prompting or fine-tuning the LLM to be more explicit about its assumptions in the layouts (e.g., viewpoints) may alleviate this problem. The intermediate layout in our two-stage generation allows for more interpretability compared to our base model stable diffusion. After diagnosing the point of failure, we give an additional request for the side view and correct object sizes to the LLM. The LLM adjusted the subsequent layout generation, which allows generating images that align with the input prompt in round 2, as shown in Fig. 9(d,e). Our method also inherits biases from the base diffusion model (Luccioni et al., 2023). Moreover, although our method can handle objects not mentioned in the in-context examples (e.g., the bear and the deer in Fig. 7), the LLM may still generate better layouts for objects mentioned in the in-context examples by referencing layout examples. Our method could also be distilled into a one-stage text-to-image diffusion model to improve its prompt understanding abilities without leveraging LLMs at inference time for the ease of deployment.

## 6    Summary

In this paper, we enhance the prompt understanding capabilities of text-to-image diffusion models. We present a novel training-free two-stage generation process that incorporates LLM-based text-grounded layout generation and layout-grounded image generation. Our method also enables instruction-based scene specification and generation from prompts in languages unsupported by the base diffusion model. Our method outperforms strong baselines in accurately following the prompts in text-to-image generation.

**Acknowledgements.** The authors would like to thank Aleksander Holynski for the helpful discussions.

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

# A    Preliminary introduction to latent diffusion models

The layout-to-image stage (i.e., the image generation stage) of our method LMD builds on off-the-shelf text-to-image Stable Diffusion models, which is based on the latent diffusion framework (Rombach et al., 2022). We present a preliminary introduction to the latent diffusion framework in this section and define the key terms used in our work. We encourage the readers to check Rombach et al. (2022) for a detailed explanation of the latent diffusion framework.

Latent diffusion models (Rombach et al., 2022) are powerful generative models that learn the data distribution of complex, high-resolution image datasets. Before training a latent diffusion model, Rombach et al. (2022) first trains an image encoder that converts an image $\mathbf{x}$ into a vector $\mathbf{z}$ in the high-dimensional *latent* space and a decoder that converts $\mathbf{z}$ back to a vector in the image space that is similar to $\mathbf{x}$ in appearance. By training and sampling a diffusion model in the latent space, latent diffusion lowers the cost of training and sampling from high-resolution diffusion models and is widely used in text-to-image generation, with Stable Diffusion as a popular model based on the latent diffusion framework. Our method improves the prompt understanding of Stable Diffusion without adapting the weights.

During training, the latent diffusion framework first maps each training image, denoted as $\mathbf{x}_0$, into latent $\mathbf{z}_0$ with the image encoder that is frozen during the diffusion training stage:

$$\mathbf{z}_0 = \mathsf{Encode}(\mathbf{x}_0) \tag{8}$$

A timestep $t$ is sampled uniformly from $\{1, ..., T\}$, where $T$ is a hyperparameter.

Noise $\boldsymbol{\epsilon}$ is then sampled from a Gaussian distribution parameterized by timestep $t$ and added to the latent $\mathbf{z}_0$ to obtain noisy latent $\mathbf{z}_t$. A neural network with parameter $\theta$ learns to predict the added noise $\boldsymbol{\epsilon}$ for the forward process by minimizing the training objective:

$$\mathcal{L} = ||\boldsymbol{\epsilon} - \boldsymbol{\epsilon}_\theta(\mathbf{z}_t, t)||^2 \tag{9}$$

The neural network described above often uses a variant of U-Net (Ronneberger et al., 2015) architecture that has attention layers (Vaswani et al., 2017), and thus is also referred to as the diffusion U-Net.

At inference time, there are many sampling methods that allow the synthesis of samples from a diffusion model trained in the fashion described above. The general intuition is to go through a *reverse* process (also called *denoising* process) in which the diffusion model $\boldsymbol{\epsilon}_\theta$ iteratively predicts a noise vector $\boldsymbol{\epsilon}_\theta(\mathbf{z}_t, t)$ from $\mathbf{z}_t$ and subtracts it to transform $\mathbf{z}_t$ into a sample $\mathbf{z}_{t-1}$ that has less noise and is closer to the distribution of the training set, with $t$ initialized as $T$ and $\mathbf{z}_T \sim \mathcal{N}(0, \mathbf{I})$. The denoised sample $\mathbf{z}_0$ resembles the clean data in the latent space.

One can use DDPM (Ho et al., 2020) to perform sampling from a noise prediction model $\boldsymbol{\epsilon}_\theta$. DDPM predicts the noise $\boldsymbol{\epsilon}$ for each of the $T$ denoising steps and then obtains $\mathbf{z}_{t-1}$ from $\mathbf{z}_t$ using this formula:

$$\mathbf{z}_{t-1} = \frac{1}{\sqrt{\alpha_t}}\Big(\mathbf{z}_t - \frac{1-\alpha_t}{\sqrt{1-\prod_{i=1}^t \alpha_i}}\boldsymbol{\epsilon}_\theta(\mathbf{z}_t, t)\Big) + \sigma_t \boldsymbol{\epsilon}_t \tag{10}$$

where $\boldsymbol{\epsilon}_t \sim \mathcal{N}(0, \mathbf{I})$, $\alpha_t$ and $\sigma_t$ are parameterized by a variance schedule $\{\beta_t \in (0,1)\}_{t=1}^T$ that controls the size of the denoising step.

Denoising diffusion implicit models (DDIM, Song et al. (2020)) are a generalization to DDPM which allows sampling with fewer iterations. DDIM applies the following update rule:

$$\mathbf{z}_{t-1} = \sqrt{\alpha_{t-1}}\Big(\frac{\mathbf{z}_t - \sqrt{1-\alpha_t}\boldsymbol{\epsilon}_\theta(\mathbf{z}_t, t)}{\sqrt{\alpha_t}}\Big) + \sigma_t \boldsymbol{\epsilon}_t \tag{11}$$

Note that DDIM shares the same training procedure with DDPM, which means we can choose to perform DDIM or DDPM for a trained diffusion model. When $\sigma_t$ is set to 0, which is the case for our setting, the

denoising becomes deterministic given $\mathbf{z}_T$. The results shown in our work are obtained with DDIM with $\sigma_t = 0$, with other faster sampling methods such as Lu et al. (2022) also applicable to our method.

Since there are many sampling methods given a trained diffusion model that are applicable in the latent diffusion framework, we denote the denoising process, such as the one in Eq. (10) and Eq. (11), as

$$\mathbf{z}_{t-1} \leftarrow \texttt{Denoise}(\mathbf{z}_t) \tag{12}$$

After getting the denoised sample $\mathbf{z}_0$, we then decode the image with an image decoder:

$$\mathbf{x}_0 = \texttt{Decode}(\mathbf{z}_0) \tag{13}$$

**Text-conditional generation through cross-attention.** The above formulation describes the unconditional generation process of latent diffusion models. Models such as Stable Diffusion take text as input and perform conditional generation. The difference between conditional and unconditional generation process involves processing the input text into text features, passing the feature tokens to diffusion U-Net, and performing classifier-free guidance (Ho & Salimans, 2022), which is described as follows.

Rather than only taking the noisy input $\mathbf{x}_t$ and timestep $t$, the conditional diffusion U-Net $\boldsymbol{\epsilon}_\theta(\mathbf{z}_t, t, \tau_\theta(\mathbf{y}))$ takes in an additional text condition $\mathbf{y}$ processed by a text encoder $\tau_\theta(\cdot)$. The text encoder is a CLIP (Radford et al., 2021) text encoder in Stable Diffusion. After $\mathbf{y}$ is tokenized by the tokenizer into discrete tokens, it is processed by a Transformer (Vaswani et al., 2017) to text features $\tau_\theta(\mathbf{y}) \in \mathbb{R}^{l \times d_\text{text}}$, where $l$ is the number of text tokens in $\mathbf{y}$ after tokenization and $d_\text{text}$ is the dimension of features.

The text features $\tau_\theta(\mathbf{y})$ are then processed by the cross-attention layers in the diffusion U-Net so that the output of the U-Net can also change depending on the text. For simplicity, we only consider one cross-attention head in this preliminary introduction and refer the readers to Rombach et al. (2022) and Vaswani et al. (2017) for details with the multi-head cross-attention used in the U-Net in the latent diffusion framework.

Specifically, each cross-attention layer linearly maps the text features $\tau_\theta(\mathbf{y})$ into key and value vectors $\mathbf{k}, \mathbf{v} \in \mathbb{R}^{l \times d_\text{attn}}$, where $d_\text{attn}$ is the attention dimension. Each cross-attention layer also takes in the flattened 2D feature from the previous layer in the U-Net and linearly maps the feature into a query vector $\mathbf{q} \in \mathbb{R}^{m \times d_\text{attn}}$ where $m$ is the dimension of the previous flattened 2D image feature.

Then, a cross-attention map $\mathbf{A}$ is computed from the query $\mathbf{q}$, key $\mathbf{k}$, and value $\mathbf{v}$ vectors, which describes the affinity from the image feature to the text token feature:

$$\mathbf{A}_{uv} = \texttt{Softmax}(\mathbf{q}_u^T \mathbf{k}_v) \tag{14}$$

where $\mathbf{q}_u$ and $\mathbf{k}_v$ are linearly transformed image feature at spatial location $u$ and text feature at token index $v$ in the prompt, respectively.

The attention map is then used for computing a weighted combination of the values $\mathbf{v}$:

$$\mathbf{o}_u = \sum_v \mathbf{A}_{uv} \mathbf{v}_v \tag{15}$$

$\mathbf{o} \in \mathbb{R}^{m \times d_\text{attn}}$ is then linearly transformed to become the output of the cross-attention layer. The residual connections and layer norms are omitted in this introduction for simplicity.

Samples are generated by classifier-free guidance to ensure alignment with text prompt $\mathbf{y}$. At training time, with a small probability, the input condition $\tau_\theta(\mathbf{y})$ is randomly replaced with a learnable null token $\tau_\varnothing$. At inference time, classifier-free guidance uses the following term $\tilde{\boldsymbol{\epsilon}}_\theta(\mathbf{x}_t, t, \tau_\theta(\mathbf{y}))$ in place of the predicted noise $\boldsymbol{\epsilon}_\theta(\mathbf{x}_t, t)$ in the update rule for unconditional generation:

$$\tilde{\boldsymbol{\epsilon}}_\theta(\mathbf{x}_t, t, \tau_\theta(\mathbf{y})) = w\boldsymbol{\epsilon}_\theta(\mathbf{x}_t, t, \tau_\theta(\mathbf{y})) + (1-w)\boldsymbol{\epsilon}_\theta(\mathbf{x}_t, t, \tau_\varnothing) \tag{16}$$

where $w$ is the strength of classifier-free guidance, set to 7.5 by default in Stable Diffusion.

---

**Algorithm 1** Layout-grounded image generation.

---

**Input:** A set of captioned bounding boxes $\{(\mathbf{b}^{(i)}, \mathbf{y}^{(i)})\}_{i=1}^{N}$. Background caption $\mathbf{y}^{(\mathrm{bg})}$.
**Output:** Image $\mathbf{x}_0$.
1: $\mathbf{z}_T \leftarrow \mathtt{SampleGaussian}(\mathbf{0}, \mathbf{I})$
2: *Per-box masked latent generation:*
3: **for** each captioned box $(\mathbf{b}^{(i)}, \mathbf{y}^{(i)})$ **do**
4:     $\mathbf{z}_T^{(i)} \leftarrow \mathbf{z}_T$
5:     $\mathbf{y}^{(i)} \leftarrow \mathtt{PromptForBox}(\mathbf{y}^{(i)}, \mathbf{y}^{(\mathrm{bg})})$
6:     **for** $t \leftarrow T$ to $1$ **do**
7:         $\mathbf{z}_t^{(i)}, A_t^{(i)} \leftarrow \mathtt{AttnControl}(\mathbf{z}_t^{(i)}, \mathbf{y}^{(i)}, \mathbf{b}^{(i)})$
8:         $\mathbf{z}_{t-1}^{(i)} \leftarrow \mathtt{Denoise}(\mathbf{z}_t^{(i)}, \mathbf{y}^{(i)})$
9:     **end for**
10:     $A^{(i)} \leftarrow \mathtt{TemporalAverage}(A_t^{(i)})$
11:     $\mathbf{m}^{(i)} \leftarrow \mathtt{SAMRefine}(A^{(i)}, \mathbf{z}_0^{(i)})$ (Optional: This could be replaced with an attention thresholding instead.)
12:     $\hat{\mathbf{z}}_t^{(i)} \leftarrow \mathbf{z}_t^{(i)} \otimes \mathbf{m}^{(i)}$
13: **end for**
14: *Composed image generation:*
15: $\mathbf{z}_T^{(\mathrm{comp})} \leftarrow \mathbf{z}_T$
16: $\mathbf{y} \leftarrow \mathtt{ComposedPrompt}((\mathbf{y}^{(i)})_{i=1}^{N}, \mathbf{y}^{(\mathrm{bg})})$
17: **for** $t \leftarrow T$ to $1$ **do**
18:     **if** $t \geq rT$ **then**
19:         $\mathbf{z}_t^{(\mathrm{comp})} \leftarrow \mathtt{LatentCompose}(\mathbf{z}_t^{(\mathrm{comp})}, \hat{\mathbf{z}}_t^{(i)}, \mathbf{m}^{(i)}) \quad \forall i$
20:         $\mathbf{z}_t^{(\mathrm{comp})} \leftarrow \mathtt{AttnTransfer}(\mathbf{z}_t^{(\mathrm{comp})}, \mathbf{y}^{(\mathrm{comp})}, (A_t^{(i)})_{i=1}^{N})$
21:     **end if**
22:     $\mathbf{z}_{t-1}^{(\mathrm{comp})} \leftarrow \mathtt{Denoise}(\mathbf{z}_t^{(\mathrm{comp})}, \mathbf{y}^{(\mathrm{comp})})$
23: **end for**
24: $\mathbf{x}_0 \leftarrow \mathtt{Decode}(\mathbf{z}_0^{(\mathrm{comp})})$

---

## B  Pseudo-code for layout-grounded image generation

We present the pseudo-code for our layout-grounding stage (stage 2) in Algorithm 1. We explain the functionality of the functions used in the pseudo-code:

1. `SampleGaussian` samples i.i.d standard Gaussian as the initial noise for the latent tensor.

2. `PromptForBox` simply sets *"[background prompt] with [box caption]"* (e.g., *"a realistic image of an indoor scene with a gray cat"*) as the denoising prompt.

3. `AttnControl` performs backward guidance to minimize the energy function Eq. (2) described in Section 3 to encourage the attention to the area within the box and discourage the attention on area outside the box. The cross-attention maps $A_t^{(i)}$ are also returned in order to allow obtaining a mask for each box.

4. `Denoise` denotes one denoising step by the diffusion model.

5. `TemporalAverage` averages the cross-attention map across the timestep dimension.

6. `SAMRefine` refines the attention map by internally decoding the latent and refining with SAM. If SAM is not enabled, we perform an attention thresholding instead.

7. `ComposedPrompt` composes the prompt for overall generation. We offer two options for the overall prompt: using the original input prompt or composing the prompt as *"[background prompt] with [box caption 1], [box caption 2], ..."*. The former one allows capturing the object as well as forground-background interactions that are not captured in the layout. The latter allows captions in languages unsupported by the diffusion model and stays robust when the caption is misleading (e.g., "neither of the apples is red"). We use the latter by default but also allow the former for fine-grained adjustments.

8. `LatentCompose` spatially composes each of the latents $\mathbf{z}^{(i)}$ with respect to the corresponding mask $\mathbf{m}^{(i)}$, replacing the content of the destination latent on the masked locations. As for the order of composition, we compose the masked latents with the largest area after masking first.

9. `AttnTransfer` performs backward guidance to minimize the energy function Eq. (7) in Section 3 to encourage the attention in overall generation within the box to be similar to the attention in per-box generation in addition to attention control.

## C  Additional features and use cases from instruction-based scene specification

As shown in Section 3.3, LMD, equipped with instruction-based scene specification, allows the user to apply follow-up instruction requests in addition to the initial prompt.

Furthermore, we demonstrate two additional use cases supported by instruction-based scene specification in Fig. C.1 without additional training.

In Fig. C.1(a), instruction-based scene specification allows the users to try out different adjustments on the same generation while preserving the overall image style and layout, facilitating fine-grained content creation.

The LLM equipped in LMD can also respond to open-ended requests and present suggestions for improving the scene. Moreover, different from instruction-based image editing methods that only take one instruction without context, our instruction-based scene specification parses the instruction in its context, allowing for more natural dialog with users. For example, in Fig. C.1(b), our method can respond to instructions with phrases such as "What are some objects that you can add to make it lively?", "undo the last edit", and "adding a small pond instead".

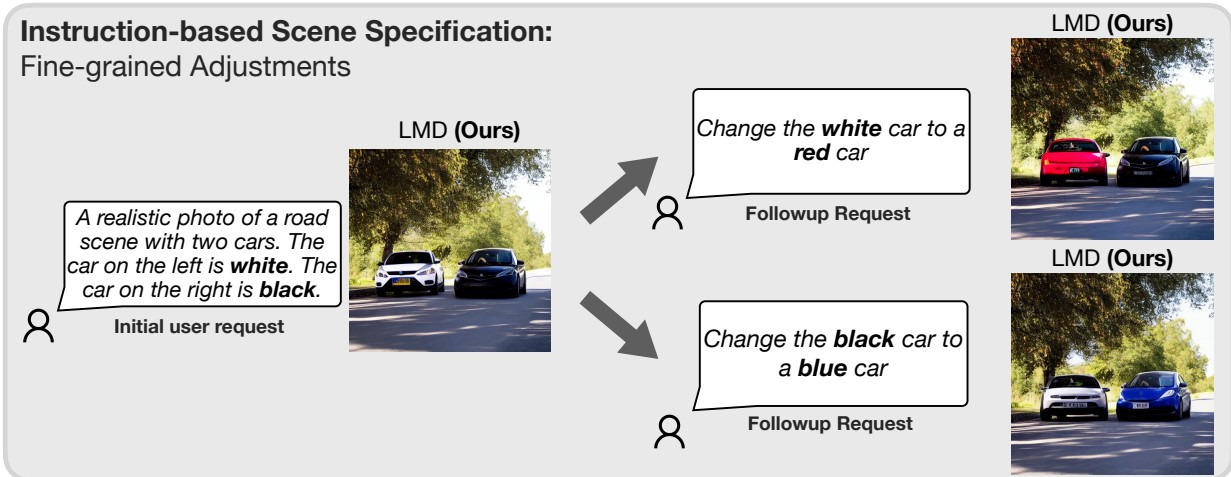

(a) LMD allows the users to try out different detailed adjustments while preserving the overall image style and layout, enabling fine-grained content creation.

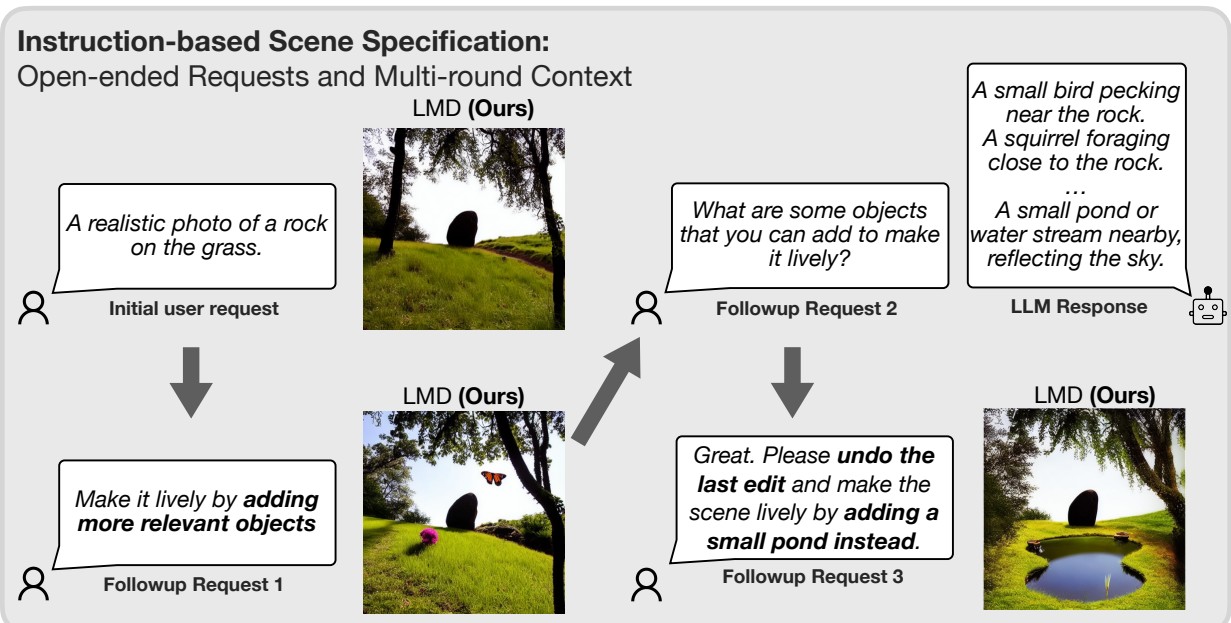

(b) The LLM used by LMD can perform open-ended scene adjustments, give suggestions, and understand user requests based on the contexts over multiple rounds of user dialog.

Figure C.1: **Additional features and use cases enabled by instruction-based scene specification.**

## D Additional ablation studies

### D.1 Text-to-layout stage

***Varying the LLM types.*** All LLMs in Table D.1 generate layouts that almost perfectly follow the requirements in the prompts, indicating the bottleneck to be the layout-to-image stage. `gpt-4` shows improved results in layout and the subsequent image generation, compared to `gpt-3.5-turbo`. The capability to generate high-quality layouts are *not limited to proprietary LLMs*, with Llama2-based `StableBeluga2` (Mahan et al., 2023; Touvron et al., 2023; Mukherjee et al., 2023) and `Mixtral-8x7B-Instruct-v0.1` (Jiang et al., 2024) also able to perform text-to-layout generation in the stage 1. We believe that fine-tuning these models will lead to even better performance in terms of text-to-layout generation.

| Stage 1 Model | Layout (Image) Accuracy Average of 4 tasks | |
|---|---|---|
| `StableBeluga2` | 96.5% | (67.0%) |
| `Mixtral-8x7B-Instruct-v0.1` | 98.3% | (77.5%) |
| `gpt-3.5-turbo` | 99.0% | (80.5%) |
| `gpt-4` | **100.0%** | **(86.3%)** |

Table D.1: **Ablations on different LLMs in stage 1.** Although proprietary models such as GPT-3.5 and GPT-4 perform the best, the ability to generate high-quality layouts is also present in open-source models Mahan et al. (2023); Jiang et al. (2024); Touvron et al. (2023). The image accuracy is benchmarked using LMD+ as stage 2.

| Stage 1 Model | Layout Accuracy Average of 4 tasks |
|---|---|
| `StableBeluga-7B` | 59.3% |
| `StableBeluga-13B` | 84.0% |
| `StableBeluga2 (70B)` | **96.5%** |

Table D.2: **Ablations on the LLM model size on StableBeluga Models (Mahan et al., 2023) based on Llama-2 (Touvron et al., 2023) for layout generation (stage 1 only).** Larger LLMs offer more accurate layout generation compared to smaller LLMs.

| Method | Image Accuracy (Average of 4 tasks) | | | |
|---|---|---|---|---|
| | $\omega = 1$ | $\omega = 2$ | $\omega = 4$ | $\omega = 8$ |
| LMD | 72.3% | 75.8% | **76.5%** | 72.5% |
| LMD+ | 79.8% | 80.0% | **80.5%** | 78.3% |

(a) Ablations on hyperparameter $\omega$.

| Method | Image Accuracy (Average of 4 tasks) | | | | |
|---|---|---|---|---|---|
| | $\lambda = 0$ | $\lambda = 1$ | $\lambda = 2$ | $\lambda = 3$ | $\lambda = 4$ |
| LMD | 70.8% | 75.0% | 76.5% | **77.3%** | 75.0% |
| LMD+ | 79.3% | 79.5% | 80.5% | **81.8%** | 78.8% |

(b) Ablations on hyperparameter $\lambda$.

Table D.3: **Ablations on hyperparameter $\omega$ and $\lambda$.** Our method is relatively stable in terms of hyperparameter values $\omega$ and $\lambda$. While we did not perform hyperparameter search, our default hyperparameter $\omega = 4$ allows optimal performance for both LMD and LMD+. For the hyperparameter $\lambda$, we found that setting $\lambda = 3$ leads to better performance compared to our default hyperparameter setting with $\lambda = 2$, which indicates that the performance of our method can be further improved through hyperparameter tuning. Underlined numbers indicate performance with our default hyperparameter selection ($\omega = 4$, $\lambda = 2$). **Bold numbers** indicate the best performance among all the hyperparameters ablated.

***Varying the LLM sizes.*** We also tested the ability of layout generation on LLMs of different model sizes. As shown in Table D.2, larger LLMs offer much better layout generation capabilities.

### D.2 Layout-to-image stage

***Varying $\omega$.*** $\omega$ is the weight for balancing the loss term on the foreground and the term on the background (Eq. (2)). While we set $\omega = 4$ by default, we ablate this design choice. As shown by the experimental results in Table D.3a, our method is relatively stable in terms of hyperparameter selection. Moreover, even though we did not perform hyperparameter search prior to determining our default hyperparameter value, our default hyperparameter $\omega = 4$ already leads to the optimal performance among the hyperparameter values that we searched in this ablation for both LMD and LMD+.

***Varying $\lambda$.*** $\lambda$ is the weight for the attention transfer term in Eq. (7). As shown in Table D.3b, we found that setting $\lambda = 3$ leads to better performance compared to our default hyperparameter setting with $\lambda = 2$, which indicates that the performance of our method can be further improved through hyperparameter tuning.

***Ablation results on T2I-CompBench (Huang et al., 2023).*** In addition to comparing our method with the baseline method Stable Diffusion in Table 6, we further combine our text-to-layout stage (stage 1) with other layout-to-image methods as stage 2 in this ablation, similar to Table 2. The results are in Table D.4, with the results for the SD baseline from Huang et al. (2023). Our method surpasses not only the base diffusion model SD but also several variants of our method that combine our stage 1 with previous layout-to-image methods as stage 2, which shows the effectiveness of our layout-grounded controller.

| Stage 1/Stage 2 | Color | Shape | Texture | Spatial |
|---|---|---|---|---|
| SD | 0.3765 | 0.3576 | 0.4156 | 0.1246 |
| LMD/MultiDiffusion | 0.4631 | 0.4497 | 0.4007 | 0.1604 |
| LMD/Backward Guidance | 0.4877 | 0.5069 | 0.4643 | 0.2361 |
| LMD/Boxdiff | 0.4579 | 0.4967 | 0.4720 | 0.1965 |
| LMD/LMD (Ours) | **0.5495** | **0.5462** | **0.5241** | **0.2570** |

Table D.4: Our method surpasses the base diffusion model SD as well as several variants of our method that combines our stage 1 with previous layout-to-image methods as stage 2 on T2I-CompBench (Huang et al., 2023).

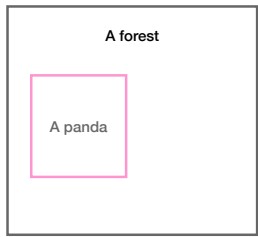 **The only in-context example with prompt** "*A panda in a forest without flowers*"

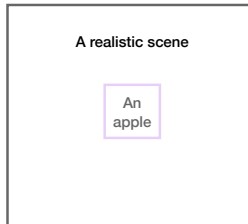 **Generated layout for prompt** "*An apple*"

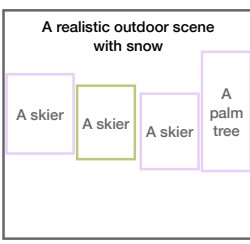 **The only in-context example with prompt** "*A realistic scene of three skiers standing in a line on the snow near a palm tree*"

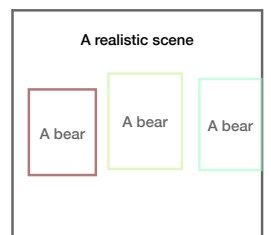 **Generated layout for prompt** "*A realistic scene of three bears*"

Figure E.1: **The generated layouts are not necessarily similar to the in-context examples in terms of the spatial distribution of boxes.** We present the LLM with *only one in-context example* and query it with a prompt that is similar to the example. **Top**: While the query and the example shares a similar structure (only one object), the LLM generates a box for "an apple" that is very different from "a panda" in terms of the size and position. **Bottom**: The LLM does not simply copy boxes for the three skiers in the in-context example to generate the boxes for three bears.

# E    Are the generated layouts distributed similarly to the in-context examples?

Since our LLM takes a few in-context examples in our text-to-layout stage, it is possible that the LLM prefers to generate samples that are similar to the in-context examples in terms of spatial distribution. To test whether this is the case, we present the LLM with only one in-context example and query it with a prompt that is similar to the example. The results are shown in Fig. E.1. Even though each of the query prompts shares a similar form to the corresponding in-context example, the LLM still generates layouts that are tailored to the objects in the query prompt (e.g., the apple and the bears) rather than copying or mimicking the layout boxes from the in-context examples. This qualitative analysis shows that even with the in-context examples as references, the LLM often generates natural layouts according to the prompts, relieving the users from heavy prompt engineering to prevent overly similar layouts between the generation and the examples.

# F    Additional visualizations

We also present Fig. F.1, which includes a qualitative comparison with Stable Diffusion v1.5 (abbreviated as SDv1) and shares the prompts with Fig. 1.

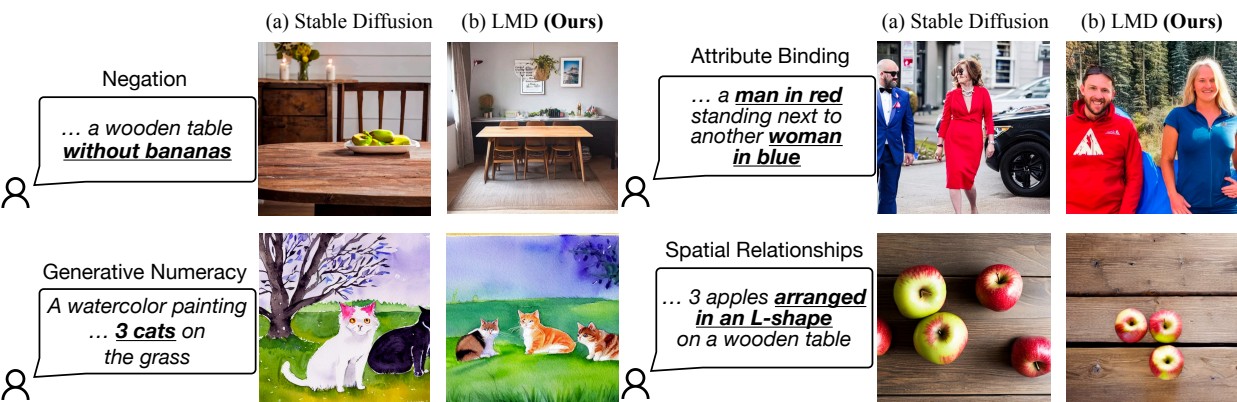

Figure F.1: **We also generate images with the same text prompts as Fig. 1 with SDv1.5 and LMD on SDv1.5.** We observe similar results which show that while Stable Diffusion Rombach et al. (2022) **(a)** often struggles to accurately follow several types of complex prompts, our method LMD **(b)** achieves enhanced prompt understanding capabilities and accurately follows these types of prompts.

## G  Benchmarking VisualChatGPT and GILL for multi-round instruction-based scene specification

VisualChatGPT (Wu et al., 2023) and GILL (Koh et al., 2023) involve LLM in their image generation pipelines and thus could potentially take instructions from multiple rounds of dialog for image generation. Therefore, in addition to the qualitative benchmark in Fig. 8, we also benchmark both methods for multi-round scene specification. As shown in Fig. G.1, the generated images quickly degrade starting from the second iteration, showing that neither method is able to take instructions from multiple rounds of dialog for image generation. In contrast, our method is able to handle several rounds of sequential requests on image generation without generation degradation, shown in Fig. 6.

## H  Details for SDXL integration

Thanks to the training-free nature of our work, our method is applicable to various diffusion models without additional training. Therefore, we also apply our method on SDXL 1.0 (Podell et al., 2023), the latest stable diffusion model which has a $3\times$ larger U-Net module compared to previous stable diffusion models (Rombach et al., 2022).

It is straightforward to apply the LMD pipeline directly to SDXL UNet, which has a very similar procedure to applying the LMD pipeline to SD v1/v2. This approach only requires marginal modifications of the LMD pipeline: different from SD v1/v2 that use only one text encoder for encoding the prompts, SDXL involves two text encoders for text feature generation, and the attention control proposed in LMD needs to be applied to the cross-attention with both text encoders taken into account. The rest follows from the standard LMD pipeline.

Inspired by methods such as Ramesh et al. (2022) that generate low resolution images and then upsample the generation to the target resolution, an alternative approach is to perform denoising with the standard LMD with a standard SDv1/v2 resolution (i.e., $512\times512$) and then perform upsampling with SDXL refiner for a few steps to the intended resolution (e.g., $1024\times1024$). Since most of the generation still happens in the standard resolution latent space, with the SDXL only involved a limited number of steps for high-resolution latents, this approach is more efficient compared to the former approach. We compare the generation for the same scene with SDXL baseline and both approaches in Fig. H.1. Both approaches present much better prompt following ability compared to SDXL baseline. We observe similar generation quality on both approaches. Therefore, we use the latter approach by default.

For Fig. 1 and Fig. 7, we use SDXL 1.0 as the base model of LMD and compare against SDXL as a strong baseline. For all other settings, including the qualitative evaluation setting, we use Stable Diffusion

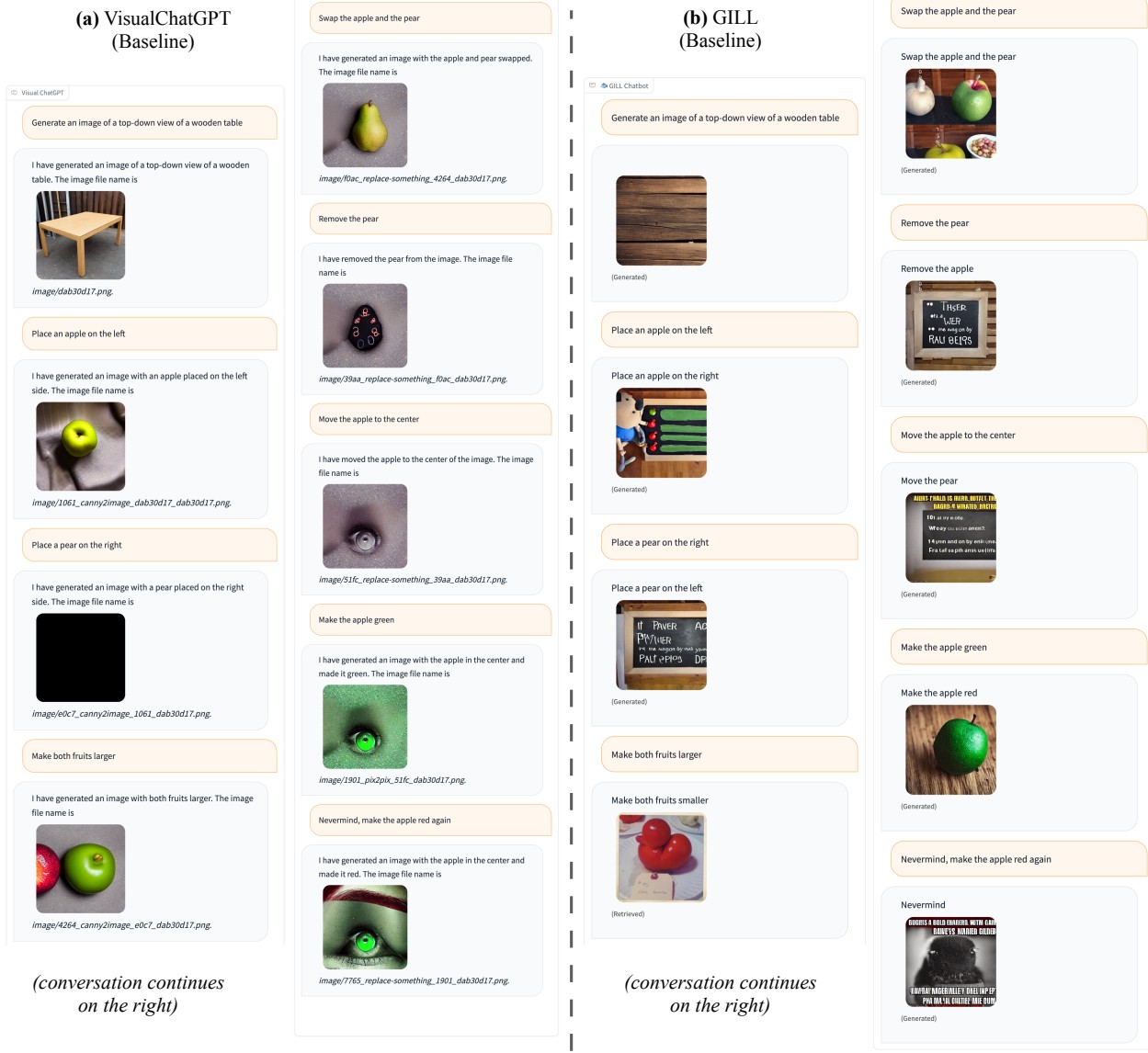

Figure G.1: VisualChatGPT Wu et al. (2023) and GILL Koh et al. (2023) generally cannot handle more than one round of image generation requests, with the generated image degraded starting from the second request. In contrast, our method is able to handle several rounds of sequential requests on image generation without generation degradation, shown in Fig. 6.

v1.5 (denoted as SDv1) unless stated otherwise. For fair comparison with Wu et al. (2023) and Koh et al. (2023) that only use Stable Diffusion v1.5, we also generate images for the same set of prompts of Fig. 1 with Stable Diffusion v1.5 in Fig. F.1.

# I    Generating images from languages not supported by the underlying diffusion model

As shown in Fig. I.1, by asking the LLM to always output layouts in English even if the prompt is non-English (e.g., Korean or Chinese as in Fig. I.1) and providing an in-context example of non-English input and English layout, LMD is able to generate images from prompts in languages not supported by the underlying

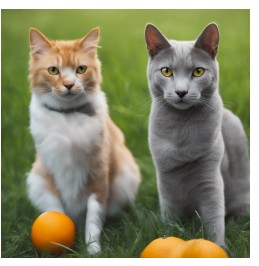
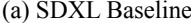

(a) SDXL Baseline

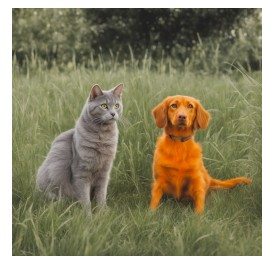

(b) Applying LMD directly on SDXL

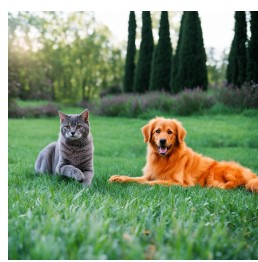

(c) Applying LMD on SD in low-res and refine with SDXL in high-res

Figure H.1: **LMD can be easily applied on the latest stable diffusion model SDXL (Podell et al., 2023).** We compare generated images from text prompt "A realistic photo of a **gray cat** and an **orange dog** on the grass". **(a)** directly generates the image from the text prompt. SDXL does not accurately generate the image from the prompt, showing that *simply scaling the diffusion model does not necessarily lead to improved prompt following ability.* **(b)** Thanks to our method being training-free, our method can be directly applied on SDXL without additional training. **(c)** An alternative way to integrate our method with SDXL is to use our method to generate low-resolution images with SD and then refine the image in high-resolution in SDXL. Since most denoising is completed in low-resolution latents, this approach is more efficient.

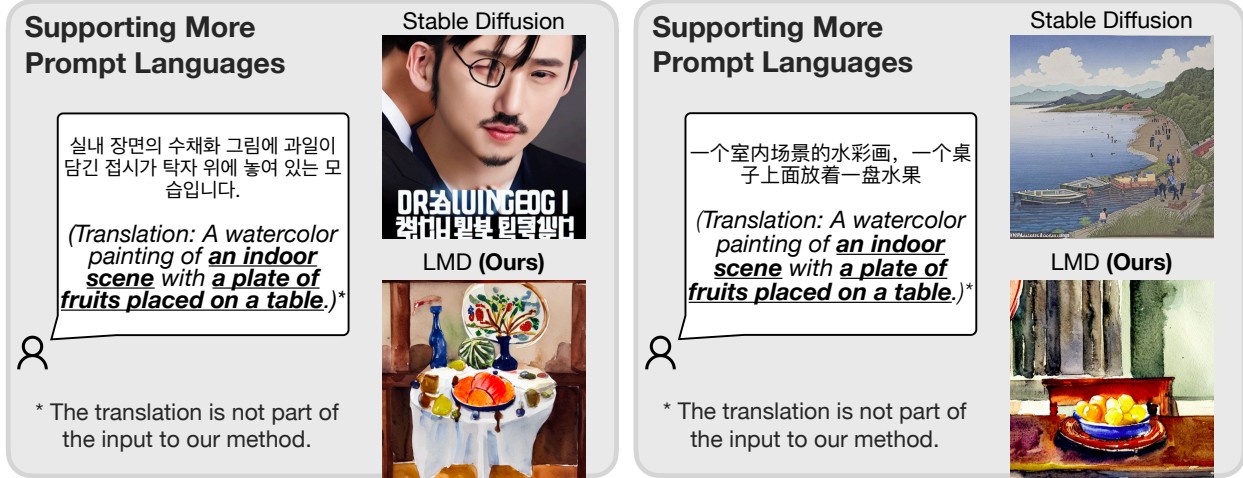

Figure I.1: By asking the LLM to always output layouts in English, **LMD is naturally able to generate images from prompts in languages not supported by the underlying diffusion model.**

diffusion model. We simply translate the prompt input of the last in-context example to non-English, while keeping the output in this example in English. No adaptation is needed on the diffusion model since the underlying diffusion model still takes in an English layout as input.

## J  Details for text-to-image benchmarks

We pick 10 common object types from the COCO dataset Lin et al. (2014) for generation[7].

For negation and generative numeracy task, each prompt requires the model to generate a layout of a scene with some number of a certain object or without a certain object. Then we count the number of objects and consider the layout to be correct if the number of the object of that particular type matches the one in the prompt, with the number ranging from 1 to 5.

---

[7]Backpack, book, bottle, bowl, car, cat, chair, cup, dog, and laptop.

The objective for each prompt in the attribute binding task is to generate an object of a color and another object of another color, for which the evaluation is similar to other tasks.

For the spatial relationship task, we generate an object at a certain location and another object at an opposite location (left/right and top/bottom). We then check the spatial coordinates of the boxes to ensure the layout exactly matches the prompt. In each task, we generate 100 text prompts, with 400 text prompts in total.

**Prompts.** For the negation benchmark, we use the prompt *A realistic photo of a scene without [object name]*.

For generative numeracy, we use the prompt *A realistic photo of a scene with [number] [object name]*.

For attribute assignment, we use the prompt *A realistic photo of a scene with [modifier 1] [object name 1] and [modifier 2] [object name 2]*, where the two modifiers are randomly chosen from a list of colors (red, orange, yellow, green, blue, purple, pink, brown, black, white, and gray).

For the spatial relationship benchmark, we use the prompt *A realistic photo of a scene with [object name 1] on the [location] and [modifier 2] [object name2] on the [opposite location]*, where the location is chosen from left, right, top, and bottom.

**Implementation details.** For LMD, we use Stable Diffusion v1.5 by default. For LMD+, we use GLIGEN (Li et al., 2023b) model without additional training or adaptation. We selected the GLIGEN (Li et al., 2023b) model trained based on Stable Diffusion v1.4, which is the latest at the time of writing. We use $\eta = 5$, $\lambda = 2.0$, $r = 0.4$, guidance scale 7.5. The energy minimization is repeated 5 times for each denoising timestep and linearly decreases for every five denoising steps until the repetition is reduced to 1, and we do not perform guidance after 30 steps. $k$ in the $\mathsf{Topk}(\cdot)$ in Eq. (2) is set to 20% of the area of the mask for each mask. The background part (second term) of Eq. (2) is weighted by $\omega = 4.0$. We run the denoising process with 50 steps by default. We only perform latent compose in the first half of the denoising process (first 25 steps). The qualitative visualizations/quantitative comparisons are generated by LMD+/LMD, respectively, by default unless stated otherwise.

## K   Our LLM prompt

Our LLM prompt is listed in Table K.1. Our in-context examples are listed in Table K.2.

```
1  You are an intelligent bounding box generator. I will provide you with a caption for a photo,
       image, or painting. Your task is to generate the bounding boxes for the objects mentioned in
       the caption, along with a background prompt describing the scene. The images are of size
       512x512. The top-left corner has coordinate [0, 0]. The bottom-right corner has coordinnate
       [512, 512]. The bounding boxes should not overlap or go beyond the image boundaries. Each
       bounding box should be in the format of (object name, [top-left x coordinate, top-left y
       coordinate, box width, box height]) and should not include more than one object. Do not put
       objects that are already provided in the bounding boxes into the background prompt. Do not
       include non-existing or excluded objects in the background prompt. Use "A realistic scene"
       as the background prompt if no background is given in the prompt. If needed, you can make
       reasonable guesses. Please refer to the example below for the desired format.
2
3  [In-context Examples]
4
5  Caption: [User Prompt]
6  Objects:
```

Table K.1: Our full prompt to the LLM for layout generation. LLM starts completion from "`Objects:`".

```
1  Caption: A realistic image of landscape scene depicting a green car parking on the left of a
       blue truck, with a red air balloon and a bird in the sky
2  Objects: [('a green car', [21, 281, 211, 159]), ('a blue truck', [269, 283, 209, 160]), ('a red
       air balloon', [66, 8, 145, 135]), ('a bird', [296, 42, 143, 100])]
3  Background prompt: A realistic landscape scene
4  Negative prompt:
5
6  Caption: A realistic top-down view of a wooden table with two apples on it
7  Objects: [('a wooden table', [20, 148, 472, 216]), ('an apple', [150, 226, 100, 100]), ('an
       apple', [280, 226, 100, 100])]
8  Background prompt: A realistic top-down view
9  Negative prompt:
10
11 Caption: A realistic scene of three skiers standing in a line on the snow near a palm tree
12 Objects: [('a skier', [5, 152, 139, 168]), ('a skier', [278, 192, 121, 158]), ('a skier', [148,
       173, 124, 155]), ('a palm tree', [404, 105, 103, 251])]
13 Background prompt: A realistic outdoor scene with snow
14 Negative prompt:
15
16 Caption: An oil painting of a pink dolphin jumping on the left of a steam boat on the sea
17 Objects: [('a steam boat', [232, 225, 257, 149]), ('a jumping pink dolphin', [21, 249, 189,
       123])]
18 Background prompt: An oil painting of the sea
19 Negative prompt:
20
21 Caption: A cute cat and an angry dog without birds
22 Objects: [('a cute cat', [51, 67, 271, 324]), ('an angry dog', [302, 119, 211, 228])]
23 Background prompt: A realistic scene
24 Negative prompt: birds
25
26 Caption: Two pandas in a forest without flowers
27 Objects: [('a panda', [30, 171, 212, 226]), ('a panda', [264, 173, 222, 221])]
28 Background prompt: A forest
29 Negative prompt: flowers
30
31 Caption: An oil painting of a living room scene without chairs with a painting mounted on the
       wall, a cabinet below the painting, and two flower vases on the cabinet
32 Objects: [('a painting', [88, 85, 335, 203]), ('a cabinet', [57, 308, 404, 201]), ('a flower
       vase', [166, 222, 92, 108]), ('a flower vase', [328, 222, 92, 108])]
33 Background prompt: An oil painting of a living room scene
34 Negative prompt: chairs
```

Table K.2: Our in-context examples. We use fixed in-context examples for layout generation.

