# OpenReview forum: "LLM-grounded Diffusion: Enhancing Prompt Understanding of Text-to-Image Diffusion Models with Large Language Models"
_TMLR — Accepted by TMLR_

### Review · Reviewer_nXop · 2023-11-22

**Summary Of Contributions:**

Current image generation models struggle to depict scenes that match rigid requirements, such as containing the right quantity or spatial ordering of elements. This work proposes an LLM-powered training-free method for doing so that leverages the LLM's ability to encode textual restrictions from the prompt. The LLM is first few-shot prompted to generate a scene layout using coordinate-based bounding boxes. Then, the diffusion model (here, variants of StableDiffusion) is carefully controlled to generate scenes that respect these layouts while generating otherwise equally natural looking images. The resulting images are both qualitatively and quantitatively found to be better aligned with instructions.

**Audience:**

Yes

**Broader Impact Concerns:**

I did not find such issues. The broader impacts of this work are roughly in line with other work on image generation and work involving LLMs.

**Claims And Evidence:**

Yes

**Requested Changes:**

In line with the weaknesses noted in the previous paragraph, please add/change:

- The study in Sec. 4.2.3, either expanding the prompt set manually rated substantially (~50 examples would be good) or removing the section entirely.
- Ablations for the $\omega$ and $\lambda$ terms to highlight the model's sensitivity to those choices.
- A discussion and/or ablations surrounding the role of the (examples in the) few-shot prompt in the model's performance.
- The discussion surrounding the benefit relative to related work, slightly. E.g., by noting the benefit of approaches that do use training as not requiring an additional LLM in the loop.
- Minor changes/additions pointed out above (e.g., fix Alg. 1).

**Strengths And Weaknesses:**

This work presents substantial progress on instruction-following capabilities in image generation. The proposed method is relatively straightforward to implement as it relies on pretrained models. The paper is both well-written and provides a rich series of analyses and comparisons that reinforce the contribution.

A few parts of the paper would benefit from additional analysis or explanation. I list the more prominent cases here followed by a list of fairly minor comments.

Since no training is involved, the approach leans heavily on a few elements that were not or nearly not ablated. This includes constants such as $\omega = 4.0$ in Eq. 3 and $\lambda=2.0$ in Eq. 8, as well as the choice of few-shot examples in the model prompt. The work would benefit from showing how changing these, especially the former 2, impacts performance. The prompt, in particular, might benefit from a discussion focusing on the limitations of encoding images as scenes involving bounding boxes. For instance, does the prompt introduce a bias towards elements that are spatially distributed similar to the examples (or where the examples intentionally chosen for diversity)? Does using fewer examples in the prompt worsen performance? One might also imagine this working relatively poorly for prompts that involve hard-to-segment scenes, such as one actor standing in front of another, a large scene with many elements (e.g., dozens of people on a beach), or unusually interconnected/abstract elements (e.g., an Escher painting).

Another issue is the evaluation in Sec. 4.2.3. While collecting human feedback is valuable, 11 responses is too low a number to establish any meaningful trends. It is also not clear how 11 responses were used for 10 prompts -- was one prompt rated twice?

Other issues:

- In a few places, the work emphasizes its merit relative to other work in ways that would benefit from adjustment. Citing the lack of a need for training as a strength, for instance. One could imagine using images generated using this work as data augmentation in order to substantially improve a SD model, not unlike DALLE-3, in which case training would result in a model that could match its abilities without requiring an LLM. In another, it mentions related work that focuses on a "limited closed set", but does not acknowledge that a few-shot prompt of examples (as used in this work) confers biases of its own towards a limited (if not closed) set of demonstrations.
- Perhaps I missed it, but I could not find how the model is prompted when modifying scenes. Presumably it is shown both the few-shot prompt and the prior turn of the conversation?
- Alg. 1 on P17 has some unnecessary line breaks (e.g. on L18/19).

---

> ### Author Response · Authors · 2023-12-29
>
> Thanks for your wonderful review and detailed comments! We updated our paper PDF to reflect changes made according to your review. Here we address each point of your comments:
>
> ## Ablations on the constants
>
> > Since no training is involved, the approach leans heavily on a few elements that were not or nearly not ablated.
> >
> > This includes constants such as $\omega=4.0$ in Eq. 3 and $\lambda=2.0$ in Eq. 8, as well as the choice of few-shot examples in the model prompt. The work would benefit from showing how changing these, especially the former 2, impacts performance.
>
> We added additional ablations in Section C in the appendix. Specifically, we present the ablations of $\omega$ in Tab. 9 (a) and the ablations of $\lambda$ in Tab. 9 (b). The ablation shows that **our method is stable with respect to these hyperparam selections**. It also suggests that **the performance of our method can be further improved with additional hyperparam search**. Thanks for the suggestions! We address your concerns about few-shot in-context examples in the response below.
>
> ## Are the generated layouts distributed similarly to the in-context examples?
>
> > Does the prompt introduce a bias towards elements that are spatially distributed similar to the examples (or where the examples intentionally chosen for diversity)?
>
> Great question! Since our LLM takes a few in-context examples in our text-to-layout stage, it is possible that the LLM prefers to generate samples that are similar to the in-context examples in terms of spatial distribution. To test whether this is the case, we present the LLM with only one in-context example and query it with a prompt that has a form similar to the given example. For instance, with the only in-context example as "A panda in a forest without flowers", we query the LLM to generate a layout for the prompt "An apple" to see whether the spatial location of the box for the apple is similar to the location of the box for the panda in the in-context example. The detailed setting and results are shown in our Appendix D and Fig. 10 (revised paper).
>
> We show that even though each of the query prompts in Fig. 10 shares a similar form to the corresponding in-context example, the LLM still generates layouts that are tailored to the objects in the query prompt (e.g., the apple) rather than copying or mimicking the layout boxes from the in-context examples. This qualitative analysis shows that **even with the in-context examples as references, the LLM often generates natural layouts according to the prompts, relieving the users from heavy prompt engineering to prevent overly similar layouts between the generation and the examples**.
>
> ## Ablations on the number of prompts
>
> > Does using fewer examples in the prompt worsen performance?
>
> We include a quantitative evaluation of the number of in-context examples (the number of shots) in Tab. 5 of the PDF after revision. We are happy to perform more ablations according to your suggestions.
>
> ## Precision and expressivity of the intermediate representations
>
> > One might also imagine this working relatively poorly for prompts that involve hard-to-segment scenes, such as one actor standing in front of another, a large scene with many elements (e.g., dozens of people on a beach), or unusually interconnected/abstract elements (e.g., an Escher painting).
>
> Indeed! Our method proposes to use an intermediate representation for improved text-to-image generation. Although we mainly discuss using 2D layout boxes as the intermediate representation, we believe that the ability to generate spatially fine-grained scenes can be further improved by using other forms of intermediate representation.
>
> For instance, one could extend the LLMs to output 2D masks by fine-tuning with a mask decoder so that the LLM is able to express fine-grained layout. Furthermore, one could also extend our work to use 3D bounding boxes as an intermediate representation for occlusion awareness.
>
> Note that since our layout-to-image stage takes both the intermediate representation layout and the user prompt text as input, our method is expected to at least perform on par with baseline text-to-image diffusion models in these challenging cases. We leave these potential extensions to future research.

---

> > ### Author Response · Authors · 2023-12-29
> >
> > ## Number of responses in evaluator-based assessment
> >
> > > Another issue is the evaluation in Sec. 4.2.3. While collecting human feedback is valuable, 11 responses is too low a number to establish any meaningful trends. It is also not clear how 11 responses were used for 10 prompts -- was one prompt rated twice?
> >
> > > The study in Sec. 4.2.3, either expanding the prompt set manually rated substantially (~50 examples would be good) or removing the section entirely.
> >
> > We use 10 responses **per evaluator**, with 11 evaluators. This leads to 110 responses for each of the two questions. We observe small variations across the responses, especially for the primary question that asks about the alignment between the text prompt and the generated image. This is mainly because most of the images generated by SD simply do not correctly follow the text prompts (e.g., not understanding left and right, as demonstrated in Fig. 1), while our method can follow these prompts well.
> >
> > As discussed in Sec 4.5, we observe significant gaps between our method and baseline SD in terms of text-image alignment (88% of the responses vote for our generation vs 10% of the responses vote for SD), which is unlikely due to randomness.
> >
> > Since most established layout-to-image methods (e.g., BoxDiff, Layout Guidance) did not conduct an evaluator-based assessment, the absence of an evaluator-based assessment should not be a reason for precluding the publication of our work. Therefore, we are also open to removing this evaluation if the reviewer believes the number of responses is too small for conclusive results.
> >
> > ## Using our method for data augmentation/distillation
> >
> > > In a few places, the work emphasizes its merit relative to other work in ways that would benefit from adjustment. Citing the lack of a need for training as a strength, for instance.
> > > One could imagine using images generated using this work as data augmentation in order to substantially improve an SD model, not unlike DALLE-3, in which case training would result in a model that could match its abilities without requiring an LLM.
> >
> > Good idea! Indeed, it is possible to distill our method into an existing text-to-image model to obtain improved prompt understanding abilities without LLMs at inference time. We have added descriptions of this extension in the discussions section and left the extension as a future work.
> >
> > ## Generalization to novel classes
> >
> > > In another, it mentions related work that focuses on a "limited closed set", but does not acknowledge that a few-shot prompt of examples (as used in this work) confers biases of its own towards a limited (if not closed) set of demonstrations.
> >
> > Good point! We clarify the differences that this statement refers to:
> >
> > Different from previous works that focus on a closed set of object classes, our method is not limited to objects in the in-context examples. **For example,** as shown in the upper right example in Fig. 7, our method is able to generate layouts for a deer and a bear, while neither appears in our in-context examples.
> >
> > Furthermore, as shown in the lower right example in Fig. 7, our method is able to output a horizontal box for the horse and a vertical box for the astronaut, despite the fact that **neither of them appears in the in-context examples**. This indicates that rather than simply treating these object nouns as generic objects without understanding what they refer to, **the LLM is actually able to provide object-specific layouts even without examples as references**.
> >
> > We do acknowledge that it is beneficial to have in-context examples portraying the target objects for the LLM to reference, and thus the generation could be more accurate for the objects mentioned in the in-context examples. However, as shown in Fig. 7, our method still achieves significant improvements for objects that are not in the in-context examples, showing our method's applicability on generating images with open-set object classes in the text prompt. **We have revised our work and added clarifications in Sec. 5.**
> >
> >
> > Thank you again for your positive review! Please feel free to share any additional questions and comments you may have so that we can promptly address them for you.

---

> ### Author Response · Authors · 2024-01-17
>
> Dear Reviewer nXop,
>
> We really appreciate your positive feedback on our work! As the author-reviewer discussion period is ending, we would like to follow up with you to see if you have any additional questions about our responses to your comments.
>
> Please don't hesitate to let us know if there is anything else that you would like to ask or comment on. We're more than willing to provide further details during the author-reviewer discussion.
>
> Thanks so much!
>
> Authors

---

### Review · Reviewer_UKCc · 2023-11-24

**Summary Of Contributions:**

Text-to-image diffusion models can fail to correctly capture the spatial relationships, absence, semantic attributes, or number of objects in a prompt. This submission presents a training-free method to guide the generation of text-to-image diffusion models with layouts generated by a large language model.

The effectiveness of the proposed method is evaluated quantitatively on two benchmarks: a novel one that includes negation, numeracy, attribute binding, and spatial relationships; and the existing T2I-CompBench. Samples are also compared qualitatively by means of a reader study. The proposed method achieves good performance across all benchmarks.

**Audience:**

Yes

**Claims And Evidence:**

Yes

**Requested Changes:**

**Technical novelty**

I appreciate the authors for citing and comparing with several recent methods that have been proposed to overcome these limitations of text-to-image diffusion models. In particular, when referring to the works by Bar-Tal et al., Chen et al., and Xie et al., the authors mention the presented submission differentiates itself by focusing on *instances* rather than *semantics*. I am not sure I fully understand this claim, could the authors expand on this?

As presented in Sec. 3.2, the submission falls within the framework proposed by Chen et al. and Xie et al. to guide diffusion by cross-attention optimization. Then, I do not understand what the difference between *semantics* and *instances* is in this context. Do the authors mean that the "masked latents" (as presented in Sec 3.2) are constructed per-instance? Is this the main differentiating factor from existing methods, i.e. the *Per-box masked latent generation* loop in Algorithm 1?

It could be helpful to precisely state the technical novelty of the proposed method rather than briefly presenting it at a high-level.

---

**What is the price for improved performance**

The proposed method achieves state-of-the-art performance across all tasks. What is the trade-off here compared to existing methods? Is the proposed method more computationally demanding because of the per-instance approach?

---

**Quantitative benchmarks**

I was wondering whether the authors could motivate the choice of comparing with MultiDiffusion, Backward Guidance, and BoxDiff on the novel benchmark only and not also on T2I.

---

**Unclear claims**

There are a few claims throughout the paper that I am not sure I understand completely:

* Sec. 3, introductory paragraph: "given text prompt y, *potentially* by denoising". What do the authors mean by "potentially" here? Are there other ways their proposed LMD method could generate images? If so, could the authors expand on this?

* Sec. 3.2: "and $V_i$ contains ...". I am not sure I understand this sentence, and in particular what tokens are included in $V_i$. Could the authors clarify this?

* Sec. 3.3: "Since we edit the layout rather than the raw pixels ... as demonstrated in Fig. 6". It is unclear to me what the difference between editing the layout and editing the "raw pixels" is. Furthermore, the sentence reads as Fig. 6 demonstrates certain editing instructions cannot be achieved by Brooks et al. However, Fig. 6 presents a case where LMD can follow editing instructions. I am not sure I understand what instructions LMD can follow that Brooks et al. cannot. Could the authors make this claim more precise?

* Sec. 4.3, Layout to image stage: "compared with training-free semantic control methods". Could the authors clearly state what methods they are referring to?

* Sec. 4.3: "Switching the base diffusion model without hyperparam tuning". What do the authors mean by "hyperparam tuning"? I was under the impression that LMD is training-free. Could the authors clarify in what capacity hyperparameter tuning is a part of their proposed method?

* Sec. 6: "Our method outperforms *strong* baselines". I am not sure whether *strong* is appropriate here. In Table 2, all competing methods do not achieve more than around 60% in the proposed tasks.

---

**Presentation**

1. Extensive use of jargon: throughout the paper, several terms are used without definition (e.g., latents, decoding, tokens, text features, hyperparam, noise-prediction network, cross-attention). These terms should be defined precisely to improve clarity.
2. Equations are not fully defined: the Denoise function in Eq. 5 and the Compose function in Eq. 7 are not defined. I think the Compose function is defined in the Appendix but there it appears with a different name. A more extensive introduction on how Stable Diffusion achieves text-to-image diffusion should be included to better frame the role of cross-attention in Eq. 1.
3. Figures are cited in the text out of order.
4. Not all Appendices are cited in the text.
5. Page 10, which contains 5 tables, significantly thwarts readability. Could the authors place tables where they are referred to in the text? Or move some tables to the appendix?
6. Typos in Algorithm 1:

Should the output be $x_0$ rather than $z^{comp}$?

Line 8: what does the sentence "with cross-attention map extracted to $A'^{(i)}_t$ mean?

Lines (18-19) and (20-21): could these pair of lines not be broken and put on one line? Also, I think $A^{(i)}$ is missing a $i=1, \dots, N$? Or am I missing something about the algorithm.

7. Tables 9 and 10 are cited in the Appendix but are missing from the text.

**Strengths And Weaknesses:**

**Strenghts**

1. Extensive empirical evaluation with several baselines across benchmarks.
2. State of the art performance across all reported metrics.

**Weaknesses**

1. Unclear technical novelty compared to existing training-free, layout-based guidance methods for text-to-image diffusion models.
2. Unclear claims.
3. Presentation could be made more precise.

I will expand on these points in the requested changes, and I am looking forward to discussing with the authors to clarify my misunderstandings.

---

> ### Author Response · Authors · 2023-12-29
>
> We really appreciate your detailed review and highly valuable suggestions! We carefully checked each of your comments and made a lot of changes according to your requests. Our paper PDF has been updated to reflect these changes. We address each of your points below:
>
> ## Technical novelty
> > I appreciate the authors for citing and comparing with several recent methods that have been proposed to overcome these limitations of text-to-image diffusion models. In particular, when referring to the works by Bar-Tal et al., Chen et al., and Xie et al., the authors mention the presented submission differentiates itself by focusing on instances rather than semantics. I am not sure I fully understand this claim, could the authors expand on this?
> >
> > As presented in Sec. 3.2, the submission falls within the framework proposed by Chen et al. and Xie et al. to guide diffusion by cross-attention optimization. Then, I do not understand what the difference between semantics and instances is in this context. Do the authors mean that the "masked latents" (as presented in Sec 3.2) are constructed per-instance? Is this the main differentiating factor from existing methods, i.e. the Per-box masked latent generation loop in Algorithm 1?
> >
> > It could be helpful to precisely state the technical novelty of the proposed method rather than briefly presenting it at a high-level.
>
> Thanks for your comment! However, we believe there are some misunderstandings in terms of the contributions of our work, which we would like to address before explaining the differences in terms of semantics and instance control.
>
> Our method focuses on improving the prompt understanding ability of a text-to-image diffusion model, as explained in the beginning of Sec. 3. The text-to-image generation setting that our method follows is different from the layout-to-image setting that previous works such as MultiDiffusion (Bar-Tal et al.), Layout Guidance (Chen et al.), and BoxDiff (Xie et al.) focus on. **These works require layouts as input and fall back to their baseline (Stable Diffusion) when used to perform text-to-image generation** (i.e., with only a global caption but no boxes in the provided layout). This is also the reason that we only compare with our true baseline Stable Diffusion in Tab. 1. However, these layout-to-image methods can be used as subroutines to create ablated variants of our methods, which we discuss in the contribution 2 section below.
>
> Here we summarize our contributions.
>
> ### Our contribution 1: our overall text-to-image pipeline
> Our first contribution is a two-stage generation pipeline that **improves the prompt understanding ability of diffusion models in a text-to-image setting**. In this proposed pipeline, we use an LLM for text-to-layout generation (stage 1) and a layout-grounded diffusion model for layout-to-image generation (stage 2).
>
> While layout-to-image methods inherently require layouts as inputs, the idea of involving an LLM in the generation pipeline allows the improvement of prompt understanding ability in a text-to-image setting **with no layouts provided**, which we believe should not be overlooked.
>
> Our proposed pipeline is flexible in terms of the selection of the LLM and the layout-grounded diffusion method, which has been extensively validated by the ablation studies in the experiments section. Specifically, **methods such as Bar-Tal et al., Chen et al., and Xie et al. can be used as the stage 2 of our pipeline, which we ablated in Tab. 2**, but these methods on their own do not perform text-to-image generation and thus are not directly comparable with our overall method.
>
> Our work is among the first works that involve LLMs in a text-to-image generation pipeline. Recent concurrent works [VisualChatGPT](https://arxiv.org/abs/2303.04671) and [GILL](https://arxiv.org/abs/2305.17216) also leverage an LLM and a text-to-image generation model (Stable Diffusion). However, in comparison with our method, as shown in Fig. 8, these methods still suffer from problems that can be successfully solved by our proposed pipeline in Fig. 1.

---

> > ### Author Response · Authors · 2023-12-29
> >
> > ### Our contribution 2: our training-free layout-conditioned generation (stage 2)
> >
> > In addition, we introduce a novel layout-to-image method called layout-grounded Stable Diffusion. This is used as the stage 2 of our pipeline to generate images grounded on bounding box layouts from the LLM. Compared to previous training-free layout-grounded generation methods, our method applies control on **the instanceness rather than semantics**. Specifically, previous methods perform layout control by only manipulating the cross-attention maps. Since the cross-attention map represents text-to-image similarities, such manipulation could only control "which part of the image corresponds to which concept", but it could not control the number of object instances in the specified region or the arrangement of object instances in regions with the same semantics. One prominent consequence is that **these methods could not distinguish between two boxes of the same concept side-by-side touching each other and one large box that is the union of the smaller boxes**.
> >
> > In contrast, our layout-grounded Stable Diffusion first generates a masked latent for each object instance, with the prompt specifically asking for one object (Sec 3.1). These masked latents are then composed together to serve as hints to guide the overall image generation (Sec 3.2), with the strength of hints adjustable to prevent artifacts in the overall generation. Since each masked latent only represents one object instance instead of a semantic concept, **our method ensures that only one object will be generated in each box**, allowing **precise control at an instance level without additional training for the first time**.
> >
> > To verify that it is advantageous to use our layout-grounded Stable Diffusion in the layout-to-image stage of our pipeline, **we perform ablations on our pipeline in Tab. 2, swapping our stage 2 with other layout-to-image methods (e.g., Bar-Tal et al., Chen et al., and Xie et al.) while keeping stage 1 the same**. Our method surpasses these previous methods that only perform coarse-grained control on a semantic level.
> >
> > ### Our contribution 3: instruction-based scene specification, broader language support, and a prompt understanding benchmark for text-to-image generative models
> >
> > We discovered that our overall proposed method LMD enables instruction-based scene specification (Fig. 6) and allows broader language support for the prompt input (Sec 3.3) without any additional training, which are useful for downstream applications. We also propose a prompt understanding benchmark to quantitatively evaluate the ability of a text-to-image generation pipeline to understand prompts that involve certain concepts (e.g., generative numeracy and negation in the prompt).
> >
> > We have revised our work, especially the contribution list at the end of Sec. 1, to make the contributions of our work clearer. We are open to any suggestions for further improvements of the clarity of our work. Thanks again for the comments!
> >
> > ## The price for improved performance
> > > The proposed method achieves state-of-the-art performance across all tasks. What is the trade-off here compared to existing methods? Is the proposed method more computationally demanding because of the per-instance approach?
> >
> > Good point! Our method is indeed more computationally demanding, since our method is a pipeline with two stages (which uses an LLM) and thus is more complicated than a single-stage text-to-image pipeline (e.g., Stable Diffusion). The main computational demand is in the LLM query part, as we need to either serve a model locally or query an API, both of which are associated with costs and latency. However, production pipelines such as [DALL-E 3](https://cdn.openai.com/papers/dall-e-3.pdf) are already using an LLM for prompt processing under the hood, which indicates that it is possible to reuse the existing LLM in such a pipeline also for layout generation, allowing the deployment of our method at scale.
> >
> > Furthermore, in the future, as more computationally efficient LLMs and serving methods are proposed, we expect our method to be much more efficient. Compared to methods that require training or external human annotation, our method can improve along with the evolvement of LLMs with marginal adaptation costs.

---

> > > ### Author Response · Authors · 2023-12-29
> > >
> > > ## Quantitative benchmarks
> > > > I was wondering whether the authors could motivate the choice of comparing with MultiDiffusion, Backward Guidance, and BoxDiff on the novel benchmark only and not also on T2I.
> > >
> > > Sorry for the confusion! Assuming T2I refers to the T2I-CompBench results in Tab. 6 (note that the table index refers to the revised PDF), we clarify our reasoning for the comparisons below.
> > >
> > > Our proposed method LMD follows a text-to-image pipeline. In contrast, methods such as MultiDiffusion (Bar-Tal et al.), Backward Guidance (Chen et al.), and BoxDiff (Xie et al.) are layout-to-image methods. **These methods on their own do not perform text-to-image generation and thus are not directly comparable with our overall method**. Without the layouts as inputs, these previous works simply fall back to vanilla Stable Diffusion. This is the reason for comparing against our base text-to-image diffusion model Stable Diffusion in Tab. 1.
> > >
> > > However, our method is flexible in terms of the selections of the method for layout-to-image generation. Therefore, **we perform ablations on various layout-to-image methods in Tab. 2**. Specifically, rather than comparing LMD with previous baseline methods in Tab. 2, we compare between our LMD (as both stage 1 and 2) and a variant of LMD that uses the same stage 1 along with other methods (e.g., Bar-Tal et al., Chen et al., and Xie et al.) as stage 2. Our method surpasses these variants in terms of the control quality.
> > >
> > > We perform all our ablations on our benchmark due to the rationale stated above in the initial submission. As requested by the reviewer, we also performed the ablation on the T2I-CompBench. The results are shown below:
> > >
> > > |#| T2I-CompBench | Color  | Shape  | Texture | Spatial |
> > > |-|---------------|--------|--------|---------|---------|
> > > 1| SDv1          | 0.3765 | 0.3576 | 0.4156  | 0.1246 |
> > > 2| LMD (stage 1) + MultiDiffusion (**new results**) | 0.4631 | 0.4497 | 0.4007 |	0.1604 |
> > > 3| LMD (stage 1) + Backward Guidance (**new results**) | 0.4877 | 0.5069 | 0.4643 | 0.2361 |
> > > 4| LMD  (stage 1) + Boxdiff (**new results**) | 0.4579 | 0.4967 | 0.4720 |0.1965 |
> > > 5| LMD (stage 1 and 2) | **0.5495** | **0.5462** | **0.5241** | **0.2570**|
> > >
> > > The variants of our method (result 2-4) that use other layout-to-diffusion methods as stage 2 outperform the baseline SDv1 (result 1), confirming the validity of our proposed text-to-layout stage 1.
> > >
> > > Moreover, **our method (result 5) outperforms baseline methods and other ablations (result 1-4) on all four tasks in T2I-CompBench**, showing the performance improvements from both our stage 1 and stage 2. We plan to add the above results to our paper.
> > >
> > > ## Unclear claims
> > > Thanks for your suggestions! We have significantly revised our work according to your comments. We address each of your points below:
> > >
> > > > Sec. 3, introductory paragraph: "given text prompt y, *potentially* by denoising". What do the authors mean by "potentially" here? Are there other ways their proposed LMD method could generate images? If so, could the authors expand on this?
> > >
> > > Sorry for the confusion! By "potentially", we meant that diffusion models are one way to achieve text-to-image generation setting, but there are other potential methods available for text-to-image generation (e.g., GANs). The word "potentially" was intended to describe the setting, not the method (since our LMD currently only explores using diffusion models as its foundation). We have revised this part in the latest revision to prevent confusion.
> > >
> > > > Sec. 3.2: "and $V_i$ contains ...". I am not sure I understand this sentence, and in particular what tokens are included in $V_i$. Could the authors clarify this?
> > >
> > > Certainly! Our prompt for per-box generation is "[background prompt] with **[box caption]**", and $V_i$ indicates the indices of tokens that correspond to "**[box caption]**" in the prompt. For example, while generating the per-box latent for a box with caption "a gray cat", $V_i$ indicates the token indices that correspond to "**a gray cat**" in "[background prompt] with **a gray cat**". We have added clarifications to our paper.

---

> > > > ### Author Response · Authors · 2023-12-29
> > > >
> > > > > Sec. 3.3: "Since we edit the layout rather than the raw pixels ... as demonstrated in Fig. 6". It is unclear to me what the difference between editing the layout and editing the "raw pixels" is. Furthermore, the sentence reads as Fig. 6 demonstrates certain editing instructions cannot be achieved by Brooks et al. However, Fig. 6 presents a case where LMD can follow editing instructions. I am not sure I understand what instructions LMD can follow that Brooks et al. cannot. Could the authors make this claim more precise?
> > > >
> > > > * **The difference between editing the layout and the raw pixels**: Editing layouts only involves changing the captions and the coordinates of the layout boxes that we previously generated with the LLM. A new image is then generated with our layout-to-image stage based on the updated layouts. The same random seed can be used to preserve the appearance and style of the objects and the scene before and after editing, as shown in Fig. 6 in our work. In contrast, editing the raw pixels requires changing the pixel values directly.
> > > > * There is a significant disadvantage of editing raw pixels compared to editing the layout boxes: It's often hard to follow spatial instructions with raw-pixel editing, since it often involves inpainting the background for operations such as "swap two objects" and "move/remove an object". If the holes are not inpainted correctly, there will be artifacts in the edited image.
> > > > * As described in the Fig. 13 and Sec. 5 of the InsturctPix2Pix paper (Brooks et al.), InsturctPix2Pix cannot perform actions that require spatial reasoning (e.g., move/swap objects). In contrast, since we edit the layout boxes rather than the raw pixels, we can accomplish these easily without considering hole inpainting, allowing us to handle user requests that SD + InstructPix2Pix cannot achieve.
> > > > * We provide a dialog of VisualChatGPT, which equips ChatGPT with Brooks et al. as a tool, in Fig. 12 (our revised paper) in our appendix. VisualChatGPT fails when given the same instructions as the ones given to our model in Fig. 6. We have added clarifications to our work.
> > > >
> > > > > Sec. 4.3, Layout to image stage: "compared with training-free semantic control methods". Could the authors clearly state what methods they are referring to?
> > > >
> > > > This phrase refers to training-free methods that perform layout-conditioned generation in terms of the semantics of the box regions (i.e., the training-free methods in Tab. 2). We have revised this phrase and decided to use "training-free layout-to-image generation methods that perform semantic-level grounding" instead, which contrasts with our method that performs instance-level grounding, as explained above in the response. We are open to your suggestions on the wording!
> > > >
> > > > > Sec. 4.3: "Switching the base diffusion model without hyperparam tuning". What do the authors mean by "hyperparam tuning"? I was under the impression that LMD is training-free. Could the authors clarify in what capacity hyperparameter tuning is a part of their proposed method?
> > > >
> > > > You are right! LMD is a training-free method. However, we still have a few "knobs" that can be tuned (e.g., $\lambda$ introduced in Eq. 7). Our hyperparameters refer to these terms. We switch the base diffusion model from SD v1.5 to SD v2.1 without changing any of these hyperparameters, yet the performance improvements are about the same on SD v1.5 and SD v2.1, which demonstrates the generalizability and robustness of our method **without any change of the values of these terms**. Note that SD v2 uses a different neural network architecture, data pipeline, and training recipe compared to SD v1, so this our generalization from SD v1.5 to v2.1 is non-trivial and indicates the potential to be applicable to other diffusion models. We are open to any word choice to refer to these terms in place of the term "hyperparameters", and we would like to hear your opinions!
> > > >
> > > > > Sec. 6: "Our method outperforms strong baselines". I am not sure whether strong is appropriate here. In Table 2, all competing methods do not achieve more than around 60% in the proposed tasks.
> > > >
> > > > We have revised this part accordingly. Thanks again for your suggestions!

---

> > > > > ### Author Response · Authors · 2023-12-29
> > > > >
> > > > > ## Presentation
> > > > > > Extensive use of jargon: throughout the paper, several terms are used without definition (e.g., latents, decoding, tokens, text features, hyperparam, noise-prediction network, cross-attention). These terms should be defined precisely to improve clarity.
> > > > >
> > > > > Thanks for pointing this out! We revised our work to clearly define these terms. Furthermore, **we have completely rewritten our Appendix A to provide preliminaries and clear definitions of our notations and terms**, including the ones mentioned in your suggestions. We added a reference at the beginning of our methods section (Sec. 3) to Appendix A in order to provide a brief introduction to the latent diffusion framework that our work is based on. We believe the addition of this preliminary significantly enhanced the clarity of our work, especially for readers without much background in diffusion models.
> > > > >
> > > > > > Equations are not fully defined: the Denoise function in Eq. 5 and the Compose function in Eq. 7 are not defined. I think the Compose function is defined in the Appendix but there it appears with a different name. A more extensive introduction on how Stable Diffusion achieves text-to-image diffusion should be included to better frame the role of cross-attention in Eq. 1.
> > > > >
> > > > > We added a brief explanation for the `Denoise` function in Eq. 5. Furthermore, the function is now clearly defined and described in the revised Appendix A. We have also fixed the naming inconsistencies for the `LatentCompose` function and added a description to the function in both the main text and the appendix.
> > > > >
> > > > > > Figures are cited in the text out of order.
> > > > >
> > > > > We double-checked our manuscript and found that Fig. 11 and 12 are indeed cited before Fig. 9. However, Fig. 11 and 12 are in the appendix and thus could not be easily moved to the main text due to space constraints. We have added notes to direct interested readers to the appendix for Fig. 11 and 12. We are also open to any suggestions!
> > > > >
> > > > > > Not all Appendices are cited in the text.
> > > > > > Page 10, which contains 5 tables, significantly thwarts readability. Could the authors place tables where they are referred to in the text? Or move some tables to the appendix?
> > > > > > Typos in Algorithm 1
> > > > >
> > > > > We have reorganized the tables and fixed the typos and references according to your request. Thanks again for the comments!
> > > > >
> > > > > > Tables 9 and 10 are cited in the Appendix but are missing from the text.
> > > > >
> > > > > Thanks for the feedback. Do you mind clarifying what "missing from the text" refers to? These two tables (with index Tables 10 and 11 in our revised paper) are positioned on the last page of our paper. We are open to any suggestions for changes in the positions of these two tables.
> > > > >
> > > > >
> > > > > We would like to thank you again for your detailed and encouraging review! Please feel free to share any additional questions and comments you may have so that we can promptly address them for you.

---

> > > > > > ### Comment · Reviewer_UKCc · 2024-01-06
> > > > > > **Thank you for your revision**
> > > > > >
> > > > > > I sincerely thank the authors for their thoughtful consideration of my concerns.
> > > > > > Overall, I find the revised version of the paper improved in its clarity and presentation.
> > > > > >
> > > > > > I only have a few minor comments about the revised version of the paper:
> > > > > > * Caption of Fig. 1: it may be more appropriate only to mention SDXL since results for SD are in Fig. 11.
> > > > > > * Consistency in formatting. Paragraphs are started in several different ways throughout the paper (e.g., with periods/without periods/mid sentence in Sec. 4.3). It might ease readers to consolidate.
> > > > > > * Typo in the box at the top of page 5: `512x512...` -> `512 \times 512...`
> > > > > > * Sec. 3.2, `Per-box masked latents` paragraph: `denoising from $z_T$ to $z_0$`. It could be useful to mention here these are latents and to refer to Appendix A since these objects have not been used in the main text before this sentence.
> > > > > > * Sec. 3.3, `Instruction-based scene specification` paragraph: `In contrast, we demonstrate that VisualChatGPT...`. This claim should be supported by a reference to Appendix F.
> > > > > > * Sec 4.1, `Comparing with other LLM-based image generators` paragraph: `to correctly generate in Fig. 1 and Fig. 11 in Appendix E`. It could be helpful for figures and tables in Appendices to have a numbering that starts with the letter of the Appendix. For example, Fig. 11 could be renamed to Fig. E.2. This will help readers locate objects in the paper, and, for example, disambiguate in the sentence above that Fig. 1 is in the main text and Fig. 11 is in Appendix E. I would suggest this for tables as well.
> > > > > > * Sec. 4.3, `Layout-to-image stage.` paragraph: `justified by the fact that our training-free controller [...] in attribute binding and spatial reasoning task`. This long sentence says that LMD provides comparable or improved performance compared to GLICEN across all tasks in the proposed benchmark. Making this sentence shorter and clearer might help bring the message home.
> > > > > > * Regarding `without hyperparam tuning`: it is too informal. `hyperparameter` should be spelled as in other parts of the submission. It might be worth enumerating which hyperparameters are included here (i.e., $\lambda$ and $\omega$? Are these the only hyperparameters as shown in Table 9?).
> > > > > > * Typo in Sec. 4.3, `Text-to-layout stage` paragraph: `are observed to different in different runs` -> `are observed to differ in different runs`?
> > > > > > * Sec. 4.5, `Setting` paragraph: `with our base image generator method [...] LMD+`. Is the "base image generator method" here LMD or LMD+? It is my understand LMD+ is built on top of GLICEN. Any reason why LMD+ was used here instead of LMD?
> > > > > >
> > > > > > Typos in Appendix A:
> > > > > > * Second paragraph, Rombach et al should be cited within parentheses.
> > > > > > * `Text-conditional generation through cross-attention` paragraph: `takes text` -> `take text`, `performs` -> `perform`, `The text features is then` -> `The text features are then`, `a query vector qin` -> `a query vector q \in`

---

> > > > > > > ### Author Response · Authors · 2024-01-17
> > > > > > > **Thank you for your positive feedback!**
> > > > > > >
> > > > > > > Thank you for your positive feedback! We carefully checked and revised our manuscript to further enhance its clarity according to your comments. We address your remaining questions and concerns below:
> > > > > > >
> > > > > > > > It could be helpful for figures and tables in Appendices to have a numbering that starts with the letter of the Appendix. For example, Fig. 11 could be renamed to Fig. E.2. This will help readers locate objects in the paper, and, for example, disambiguate in the sentence above that Fig. 1 is in the main text and Fig. 11 is in Appendix E. I would suggest this for tables as well.
> > > > > > >
> > > > > > > Great suggestion! We will definitely take your advice. To prevent mismatches between the figure indices in the PDF and those in our responses to other reviewers, we will update the numbering as you suggested in the final version of our work.
> > > > > > >
> > > > > > > > It might be worth enumerating which hyperparameters are included here (i.e., $\lambda$ and $\omega$? Are these the only hyperparameters as shown in Table 9?).
> > > > > > >
> > > > > > > In addition to $\lambda$ and $\omega$, which are the hyperparameters introduced by our proposed method, our method also inherits hyperparameters from the latent diffusion framework. These hyperparameters, such as the number of denoising steps, are also unchanged. We have added a footnote in Sec. 4.3 for clarifications.
> > > > > > >
> > > > > > > > Sec. 4.5, Setting paragraph: with our base image generator method [...] LMD+. Is the "base image generator method" here LMD or LMD+? It is my understand LMD+ is built on top of GLIGEN. Any reason why LMD+ was used here instead of LMD?
> > > > > > >
> > > > > > > Sorry for the confusion. We use the phrase "base image generator method" to indicate Stable Diffusion (SD), which is the diffusion model used by our method LMD/LMD+ under the hood. We have revised this part to improve the clarity.
> > > > > > >
> > > > > > > LMD and LMD+ are two variants of our proposed method, with LMD+ the better-performing one. Indeed, LMD+ not only includes all the features of LMD (LLM for layout generation, attention-based instance-level control, etc.) but also incorporates the layout conditioning mechanism from GLIGEN. Therefore, while the base framework of LMD is SD, the base framework of LMD+ is technically GLIGEN, which itself is also built on SD.
> > > > > > >
> > > > > > > However, in our text-to-image generation setting where the input is only text (without any user-provided bounding boxes), the conditioning scheme of GLIGEN simply becomes degenerate. Thus, we believe that it is reasonable to compare LMD+ with SD directly instead. We have added a footnote to our manuscript to clarify this. While we have already compared SD and LMD in various other benchmarks, we are also open to conducting an evaluator-based comparison between SD and LMD if deemed necessary by the reviewer. Thanks for pointing this detail out!
> > > > > > >
> > > > > > >
> > > > > > > We have updated our manuscript to incorporate the changes suggested in your comments, including fixes for typos and formatting issues. Thank you once again for your constructive review! We are also happy to address any remaining concerns and answer any questions you may have.

---

### Review · Reviewer_Yihf · 2023-12-21

**Summary Of Contributions:**

The paper presents a novel two-stage generation process to enhance the prompt understanding capabilities of text-to-image diffusion models. This method, called LLM-grounded Diffusion (LMD), leverages a pretrained large language model (LLM) for generating scene layouts based on text prompts. These layouts guide an existing diffusion model to produce accurate images. Key contributions include:

- Improved prompt understanding in text-to-image models using a training-free, two-stage generation process involving LLMs.
- Introduction of a novel controller to steer diffusion models based on LLM-generated bounding box layouts.
- Capability for instruction-based scene specification and support for multiple languages.

**Audience:**

Yes

**Claims And Evidence:**

Yes

**Requested Changes:**

I found the concept of instruction-based specification, as discussed in this paper, to be particularly fascinating and a standout feature. I think it would be beneficial to focus more on this aspect, especially by highlighting its practical applications and the real-world impact it can have.

**Strengths And Weaknesses:**

Strength

- Two-Stage Training-Free Method: This approach is a key strength, enabling the model to leverage pretrained components without the need for additional training. This enhances efficiency and reduces the computational resources required for deployment.
- Instruction-Based Editing: The paper introduces a method that can interpret complex instructions for image generation. This flexibility allows for more precise and creative control over the output, catering to a wide range of applications.
- Utilization of Powerful LLM Models: By integrating large language models, the paper harnesses their advanced natural language understanding capabilities. This integration significantly improves the model's ability to interpret and respond to text prompts, leading to more accurate and contextually relevant image generation.

Weakness

A major concern regarding this paper arises from the methodology illustrated in Figure 5, which may be perceived as relatively straightforward by researchers familiar with diffusion models. This simplicity risks the method being classified as somewhat trivial. Except for this concern, I believe other issues are not critically detrimental, as they also represent the inherent challenges intertwined with the paper's strengths. Below are some of such weaknesses.

- Dependence on Pretrained Models: Relying heavily on large language models for generating scene layouts might limit the model's adaptability to contexts or nuances not well-represented in the LLM's training data.
- Computational Resources: While the method is training-free, the use of powerful LLMs and diffusion models might still require significant computational resources, potentially limiting its accessibility for users with less powerful hardware.
- Generalization and Robustness: The paper's approach might face challenges in generalizing across diverse and complex scenarios, especially in cases where the text prompts are ambiguous or highly creative.

---

> ### Author Response · Authors · 2023-12-29
>
> Thank you for your encouraging review! We first address the most critical concern of the reviewer:
>
> ## Novelty of layout-to-image stage of our method
>
> > A major concern regarding this paper arises from the methodology illustrated in Figure 5, which may be perceived as relatively straightforward by researchers familiar with diffusion models. This simplicity risks the method being classified as somewhat trivial.
> >
> > Except for this concern, I believe other issues are not critically detrimental.
>
> Our method is a two-stage generation pipeline with an LLM for text-to-layout generation (stage 1) and a layout-grounded diffusion model for layout-to-image generation (stage 2). With the contributions of the overall pipeline and the use of LLM acknowledged by the reviewer as our strength, we focus on addressing the concern about the novelty of the second stage illustrated in Fig. 5 of our paper.
>
> Indeed, there are previous layout-to-image methods that manipulate the attention maps to guide the diffusion models to generate images according to box layouts, which we cited as related works. However, since the cross-attention map represents text-to-image similarities, such manipulation could only control "which part of the image corresponds to which concept", but it could not control the number of object instances in the specified region. One prominent consequence is that **these methods could not distinguish between two boxes of the same concept side-by-side touching each other and one large box that is the union of the smaller boxes**.
>
> In contrast, our layout-grounded Stable Diffusion first generates a masked latent for each object instance, with the prompt specifically asking for one object (Sec 3.1). These masked latents are then composed together to serve as hints to guide the overall image generation (Sec 3.2), with the strength of hints adjustable to prevent artifacts in the overall generation. Since each masked latent only represents one object instance instead of a semantic concept, **our simple yet effective method ensures that only one object will be generated in each box**, allowing **precise control at an instance level without additional training for the first time**.
>
> To verify that it is advantageous to use our layout-grounded Stable Diffusion in the layout-to-image stage of our pipeline, **we perform ablations on our pipeline in Tab. 2, swapping our stage 2 with other layout-to-image methods (e.g., Bar-Tal et al., Chen et al., and Xie et al.) while keeping stage 1 the same**. Our method surpasses these previous methods that only perform coarse-grained control on a semantic level.
>
> Therefore, our stage 2 method is simple and easy to understand, yet **it enables instance-level control that previous training-free methods could not achieve**, which indicates non-trivial novelty compared to previous works.
>
> We then address each point of your remaining comments:
>
> ## Dependence on pretrained models
>
> > Dependence on Pretrained Models: Relying heavily on large language models for generating scene layouts might limit the model's adaptability to contexts or nuances not well-represented in the LLM's training data.
>
> Good point! Although the LLM performs surprisingly well in most of the cases that we explored, it is possible that there are some concepts or scenes that are too out-of-distribution for the LLM to generate layouts on. Fine-tuning the LLM (potentially through public or self-hosted fine-tuning services) is one potential solution to this issue.
>
> ## Computational resources
>
> > Computational Resources: While the method is training-free, the use of powerful LLMs and diffusion models might still require significant computational resources, potentially limiting its accessibility for users with less powerful hardware.
>
> Our method is indeed more computationally demanding, since our method is a pipeline with two stages (which uses an LLM) and thus takes more resources compared to a single stage text-to-image pipeline (Stable Diffusion).
>
> The main computational cost lies in the LLM query part, as we need to either serve a model locally or query through an API. Although public LLM inference APIs are available, the call to the LLM incurs additional costs and latency for users. However, production pipelines such as [DALL-E 3](https://cdn.openai.com/papers/dall-e-3.pdf) are already using an LLM for prompt processing under the hood, which indicates that it is possible to reuse the existing LLM in the pipeline also for layout generation, allowing the deployment of our method at scale.
>
> Moreover, in the future, as more computationally efficient LLMs and serving methods are being released, we expect our method to become much more efficient.
>
> Finally, we empirically show that our method applies to various LLMs and diffusion models off-the-shelf, which **expands the applicability of our method to users that do not have the budget or hardware to collect annotated images and fine-tune a diffusion model or an LLM on their own**.

---

> ### Author Response · Authors · 2023-12-29
>
> ## Generalization and robustness
>
> > Generalization and Robustness: The paper's approach might face challenges in generalizing across diverse and complex scenarios, especially in cases where the text prompts are ambiguous or highly creative.
>
> Indeed! Our method may still face challenges when the prompt is extremely complicated. Perhaps surprisingly, the bottleneck is often not the LLM but the diffusion models (i.e., the generated layout is correct but the final image does not align with the prompt, as shown in Sec. 5). Since our method is training-free, we expect the generalization capabilities and the robustness of our method to improve with along with the release of better LLMs and diffusion models. We also believe a finer-grained intermediate representation compared to 2D bounding boxes (e.g., masks or 3D bounding boxes) will further improve the image-text alignment of our method. We leave the exploration to future research.
>
> ## Highlighting instruction-based specification
>
> > I found the concept of instruction-based specification, as discussed in this paper, to be particularly fascinating and a standout feature. I think it would be beneficial to focus more on this aspect, especially by highlighting its practical applications and the real-world impact it can have.
>
> Thanks for your acknowledgments! We will add more examples and discussions to demonstrate the use cases of our multi-round instruction-based specification in the final version of our paper.
>
>
> We would like to thank you again for your constructive review and acknowledgments! Please feel free to share any additional questions and comments you may have so that we can promptly address them for you.

---

> ### Author Response · Authors · 2024-01-17
>
> Dear Reviewer Yihf,
>
> We really appreciate your constructive feedback on our work! As the author-reviewer discussion period is coming to a close, we want to ensure that we have fully addressed your concerns. Have our responses to your comments been satisfactory, or do you have any further questions?
>
> Please don't hesitate to let us know if there is anything else that you would like to ask or comment on. We're more than willing to provide additional information during the author-reviewer discussion.
>
> Thanks so much!
>
> Authors

---

### Author Response · Authors · 2023-12-29
**General Response**

We would like to thank all the reviewers for their thoughtful and overall positive reviews as well as encouraging feedback. We are especially glad that the reviewers believe that our method's two-stage training-free approach is "a key strength" (Reviewer Yihf), our evaluation is "extensive" with our method achieving "state of the art performance across all reported metrics" (Reviewer UKCc), and our work presents "substantial progress on instruction-following capabilities in image generation" (Reviewer nXop).

We would also like to thank the reviewers for all the insightful suggestions. **We present several updates to improve the clarity of the presentation of our method as well as additional key experimental results that further strengthen our work.** We kindly remind the reviewers to refer to our updated paper PDF.

Specifically, we revised these parts of our work according to the suggestions of the reviewers:
* Sec. 1: We revised the contribution list for clarity about the contributions of our work.
* Sec. 2: We clarified the distinctions between our method and previous methods that take box conditions of a closed set of class labels.
* Sec. 3: We performed significant revisions on the equations, the notations, the definition of terms introduced in our work, and the presentation of our method to improve the clarity of this section.
* Sec. 4: We re-ordered this section and added descriptions of the new ablations requested by the reviewers.
* Sec. 5: We added discussions of several questions and potential future research directions mentioned by the reviewers.
* Appendices: We rewrote the preliminary section (Appendix A) in the appendix to introduce the readers to our setting and the diffusion-related concepts used by our work. We also fixed discrepancies and revised the descriptions of the functions used in the pseudo-code (Alg. 1 and Appendix B) for clarity. Finally, we added several key ablation studies and experiments requested by the reviewers in Appendix C and D.
* We also fixed typos and other minor parts that are unclear in the initial submission.

We respond to each of the questions and comments in the individual response. We are more than happy to discuss with the reviewers to address any additional questions.

---

### Decision · Action_Editor_sQMS · 2024-02-08

**Recommendation:** Accept with minor revision

**Comment:**

This paper address an important issue of diffusion based text-to-image generative models, where the generative image cannot well reflect the prompts. It introduce a two stage method by integrating LLM to generate object layouts. After generating the object layouts, a new algorithm is proposed to integrate multiple objects into the generation process.

All reviewers find the paper novel and interesting. There were some concerns regarding the presentation and experimental results. The authors provided detailed rebuttals and some extra experimental results to support the rebuttal. All reviewers appreciate the rebuttal and recommend acceptance.

In summary, this paper proposes a new training free method to address the inconsistency of generative images and prompts and achieve state of the art results. The impact could be signifiant. However, I highly recommend the authors to revise the current submission according to the reviews and add additional results into the revision as needed.

**Audience:**

Research in generative AI or specifically diffusion models can find the work interesting.

**Claims And Evidence:**

Yes, the claims are generally supported by experiments results.